# Revisit, Extend, and Enhance Hessian-Free Influence Functions

## Abstract

Influence functions serve as crucial tools for assessing sample influence. By employing the first-order Taylor extension, sample influence can be estimated without the need for expensive model retraining. However, applying influence functions directly to deep models presents challenges, primarily due to the non-convex nature of the loss function and the large size of model parameters. This difficulty not only makes computing the inverse of the Hessian matrix costly but also renders it non-existent in some cases. Various approaches, including matrix decomposition, have been explored to expedite and approximate the inversion of the Hessian matrix, with the aim of making influence functions applicable to deep models. In this paper, we revisit a specific, albeit naive, yet effective approximation method known as TracIn, and simplify it further, introducing the name Inner Product (IP). This method substitutes the inverse of the Hessian matrix with an identity matrix. We offer deeper insights into why this straightforward approximation method is effective. Furthermore, we extend its applications beyond measuring model utility to include considerations of fairness and robustness. Finally, we enhance IP through an ensemble strategy. To validate its effectiveness, we conduct experiments on synthetic data and extensive evaluations on noisy label detection, sample selection for large language model fine-tuning, and defense against adversarial attacks.

## 1 Introduction

Data-centric learning is a growing research field that focuses on enhancing machine learning model performance by refining the quality and characteristics of training data (Oala et al., 2023). In contrast to model-centric approaches, which prioritize improving algorithms or optimization techniques without altering the dataset, data-centric learning involves adjusting the dataset itself—through methods like trimming, relabeling, and reweighting—while keeping the learning algorithm fixed. This approach plays a vital role in areas such as model interpretability, selecting training subsets, generating synthetic data, detecting noisy labels, improving active learning, and promoting fairness in machine learning models (Chhabra et al., 2024; Kwon et al., 2023).

Sample influence estimation, as the foundation of data-centric learning, can be generally categorized into two categories (Hammoudeh & Lowd, 2022). (a) Retraining-based methods assess the sample influence by retraining the model with and without a specific sample and checking the performance change, which include the classical leave-one-out influence approach (Cook & Weisberg, 1982) and Shapley value approaches (Ghorbani & Zou, 2019; Jia et al., 2019; Kwon & Zou, 2022; Jia et al., 2018). (b) Gradient-based methods estimate influence without expensive overheads of retraining, known as influence functions. The seminal work in this category is that of (Koh & Liang, 2017), which utilizes a Taylor-series approximation and LiSSA optimization (Agarwal et al., 2017) to compute sample influences. However, the limiting assumption is that the model and loss function are convex. Despite debates on the necessity of convexity (Bae et al., 2022; Grosse et al., 2023; Basu et al., 2020; Epifano et al., 2023), challenges persist when directly applying gradient-based methods to large models. The size of model parameters complicates calculations, particularly in obtaining the inverse of the Hessian matrix. Efforts, including matrix decomposition techniques (Koh & Liang, 2017; Grosse et al., 2023; Kwon et al., 2023), aim to expedite and approximate Hessian matrix inversion, therefore enhance the feasibility of influence functions for deep learning models.

**Contributions**. In this paper, we focus on the influence function category and revisit a specific naive yet aggressive approximation method, TracIn (Pruthi et al., 2020). This method substitutes the inverse of the Hessian matrix with an identity matrix, representing it as a Hessian-free influence function—the inner product of the gradient of the validation set and a sample to be assessed, which we refer to as Inner Product (IP). We summarize our major contributions as follows:

- We simplify the checkpoints of TracIn, define it into IP formulation, and delve into the rationale behind this simple approximation, offering insights into why it performs well in practice.
- Expanding our IP framework, we extend its applicability beyond measuring sample influence on model utility to encompass considerations of fairness and robustness.
- To enhance the generalization, we propose IP Ensemble, a novel approach leveraging dropout mechanisms to simulate diverse models. IP Ensemble amalgamates IP scores from these varied models, thus increasing the method's generalization capabilities.
- By assessing the sample influence, model performance can be enhanced by trimming detrimental samples from the training or fine-tuning set, downweighting them, or even relabeling them. Specifically, we validate the effectiveness of IP through synthetic data experiments and conduct extensive real-world evaluations using IP Ensemble. These experiments span various applications, including noisy label correction for vision data, data curation aimed at fine-tuning fairer NLP models, and defense strategies against adaptive evasion adversaries.

## 2 RELATED WORK

In this section, we introduce the literature on influence functions, with a focus on the acceleration of the calculation of the inverse of Hessian matrix, followed by various applications and miscellaneous.

**Efficient Influence Estimation.** Influence functions serve as crucial tools for estimating the individual valuation of data without requiring model retraining. However, the computation of the inverse of the Hessian matrix poses challenges for large-scale data and models. To address this issue, various approaches have been proposed to simplify or estimate the inverse of the Hessian matrix effectively. A seminal work is that of (Koh & Liang, 2017), which employs a Taylor-series approximation and LiSSA optimization (Agarwal et al., 2017) to compute sample influences. Arnoldi (Schioppa et al., 2022) employs the random projection and simplified Hessian matrix for acceleration. EKFAC (Grosse et al., 2023) enhances Kronecker-Factored eigendecomposition for a precise Hessian approximation. More recently, DataInf (Kwon et al., 2023) efficiently computes influence even for large models by replacing the inverse Hessian computation with a readily computable closed-form expression, although their framework may suffer from significant theoretical errors. TracInc (Pruthi et al., 2020), a straightforward yet aggressive approximation, substitutes the inverse of the Hessian matrix with an identity matrix, essentially considering gradients directly as a measure of influence. Beyond the conventional influence function that gauges sample influence on the validation set, self-influence (Bejan et al., 2023; Thakkar et al., 2023) computes influence using the training set alone. Moving beyond using a single model checkpoint, GEX (Kim et al., 2024) leverages a geometric ensemble of multiple checkpoints to approximate influence functions, alleviating the bilinear constraint and non-linear losses. Moreover, TDA (Bae et al., 2024) also introduces a checkpoint-based segmentation approach, combining implicit differentiation and unrolling by using EKFAC (Grosse et al., 2023).

**Various Applications of Influence Functions**. With the above efficient approximation, influence functions have diverse applications. One major application is identifying detrimental samples (Hammoudeh & Lowd, 2024). The learning performance can be further improved by removing (Chhabra et al., 2024), relabeling (Kong et al., 2021), or reweighting (Thakkar et al., 2023) the identified detrimental samples, which has significant implications in various fields such as noisy label detection (Wang et al., 2020), subset selection (Ting & Brochu, 2018), and the identification of the most influential samples (Sharchilev et al., 2018; Xia et al., 2024). Other applications encompass few-shot learning (Park et al., 2021), where influence functions help improve model performance with minimal data, and recommendation systems (Li et al., 2023; Zhang et al., 2023), enhancing the accuracy and personalization of recommendations. Influence functions are also valuable in selecting data for active learning (Liu et al., 2021), fairness machine learning (Li & Liu, 2022; Wang et al., 2022; 2024), adversarial attack (Cohen et al., 2020), graph machine learning (Chen et al., 2023; Wu et al., 2023), machine unlearning (Xu et al., 2024; Tarun et al., 2023), out-of-distribution generalization (Ye et al., 2021), data privacy (Carey et al., 2023), domain adaptation (Zhang et al., 2022), to name a few.

**Miscellaneous**. Several studies have examined the fragility of influence functions in explaining deep learning model predictions. Bae et al. (2022) discovers that while influence estimates may not perfectly align with leave-one-out retraining, they approximate the proximal Bregman response function, offering valuable insights for identifying influential or mislabeled examples. Basu et al. (2020) demonstrates that the effectiveness of influence functions in neural networks varies with network architecture, depth, width, parameterization, and regularization, underscoring their fragility in deep learning due to non-convex loss functions. Epifano et al. (2023) suggests that the instability of current validation procedures, rather than non-convexity or lack of regularization, may be responsible for their unreliability. Koh et al. (2019) expands influence functions from estimating the effects of removing one point to large groups of training samples; Lyu et al. (2023) enhance influence estimation in large-scale models by concentrating on target parameters and addressing computational instability with a robust inverse-Hessian-vector product approximation; Chen et al. (2020) extend traditional influence functions to monitor the impact of pre-training data on fine-tuned model predictions, facilitating the identification of crucial pre-training examples.

## 3 METHODS

In this section, we introduce the preliminaries of the influence function, with a focus on the Hessian-free approximation, then elaborate on our extension and upgrade.

**Revisit and Simplify**. Given a training set $T=\{z_i=(x_i,y_i)\}_{i=1}^n$ and a classifier with empirical risk minimization by a convex loss function $\ell$, the optimal parameters of the classifier can be obtained by $\hat{\theta} = \arg\min_{\theta\in\Theta} \frac{1}{n}\sum_{i=1}^n \ell(z_i;\theta)$. To measure the influence of an individual data sample, we can train the model with and without the specific sample and see the performance change. However, the retrain-based approach is expensive for large-scale data and models. To avoid model retraining, influence functions estimate the effect of changing an infinitesimal weight of samples on a validation set $V=\{z_j=(x_j,y_j)\}_{j=1}^{n'}$, based on an impact function $f$ evaluating the quantity of interest. Considering the sample impact on model utility, *i.e.,* the loss on the validation set, by removing this sample from the original training set, the sample influence can be estimated as follows (Koh & Liang, 2017):

$$\mathcal{I}^{\text{util}}(-z_i) = \sum_{z_j \in V} \nabla_{\hat{\theta}}\ell(z_j;\hat{\theta})^\top \mathbf{H}_{\hat{\theta}}^{-1} \nabla_{\hat{\theta}}\ell(z_i;\hat{\theta}), \tag{1}$$

where $\mathbf{H}_{\hat{\theta}}=\sum_{i=1}^n \nabla_{\hat{\theta}}^2 \ell(z_i;\hat{\theta})$ is the Hessian matrix of the convex $\ell$ loss function.

Influence functions encounter a challenge in direct application to deep models, primarily due to the non-convex nature of the loss function and the considerable size of model parameters. This obstacle not only renders the calculation of the inverse of the Hessian matrix costly but also leads to its non-existence. Various attempts, including matrix decomposition methods (Koh & Liang, 2017; Grosse et al., 2023; Kwon et al., 2023), have been undertaken to expedite and approximate the inversion of the Hessian matrix, aiming to render influence functions viable for deep models. In this paper, we revisit a particular naive yet aggressive approximation method TracIn (Pruthi et al., 2020) by substituting the inverse of the Hessian matrix with an identity matrix, outlined as follows:

$$\mathcal{I}_{\text{IP}}^{\text{util}}(-z_i) = \nabla v^{\text{util}} \nabla_{\hat{\theta}}\ell(z_i;\hat{\theta}), \text{ and } \nabla v^{\text{util}} = \sum_{z_j \in V} \nabla_{\hat{\theta}}\ell(z_j;\hat{\theta})^\top. \tag{2}$$

The above equation is the inner product of $\nabla v^{\text{util}}$ and the sample gradient $\nabla_{\hat{\theta}}\ell(z_i;\hat{\theta})$; therefore, we call this method *Inner Product (IP)*. Note that TracIn incorporates multiple checkpoints to record model parameters throughout the optimization process, whereas IP only takes into account the final or converged model. We believe that sample influence should be assessed using a fixed model; simply summing sample influences at different stages fails to capture the dynamics of model optimization. The converged model can significantly diverge from its earlier stages, making the sample influences derived from initial checkpoints less accurate.

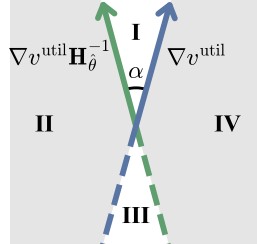

Figure 1: Illustration of $\nabla v^{\text{util}}\mathbf{H}_{\hat{\theta}}^{-1}$ and $\nabla v^{\text{util}}$.

In the following, we link the IP to influence functions and provide our insights of IP on why such a naive approximation works well in practice. Let $\nabla v^{\text{util}}=\sum_{z_j \in V} \nabla_{\hat{\theta}}\ell(z_j;\hat{\theta})^\top$, then influence functions in Eqs. (1) and (2) can be reformulated as $\nabla v^{\text{util}}\mathbf{H}_{\hat{\theta}}^{-1}\nabla_{\hat{\theta}}\ell(z_i;\hat{\theta})$ and $\nabla v^{\text{util}}\nabla_{\hat{\theta}}\ell(z_i;\hat{\theta})$. We illustrate the directions of $\nabla v^{\text{util}}\mathbf{H}_{\hat{\theta}}^{-1}$ and $\nabla v^{\text{util}}$

in Figure 1, with the influence scores roughly representing the angle between the sample gradient and $\nabla v^{\text{util}} \mathbf{H}_{\hat{\theta}}^{-1}$ or $\nabla v^{\text{util}}$. Remarkably, although influence scores by Eqs. (1) and (2) may differ, they exhibit order-consistency in several scenarios. For instance, if two samples $z_i$ and $z_{i'}$ have gradients in Region II or IV, then $\mathcal{I}^{\text{util}}(-z_i) \gtreqless \mathcal{I}^{\text{util}}(-z_{i'})$ implies $\mathcal{I}_{\text{IP}}^{\text{util}}(-z_i) \gtreqless \mathcal{I}_{\text{IP}}^{\text{util}}(-z_{i'})$. However, this order-consistency does not hold in Regions I and III. Fortunately, samples in these regions are jointly recognized by $\mathcal{I}^{\text{util}}$ and $\mathcal{I}_{\text{IP}}^{\text{util}}$ as either beneficial or detrimental, resulting in minimal practical differences. A complete analysis can be found in Appendix A.

In general, IP emerges as a straightforward, efficient, and remarkably effective alternative to influence functions. In non-convex scenarios, approximations of the inverse Hessian matrix may introduce significant errors, rendering $\nabla v^{\text{util}} \mathbf{H}_{\hat{\theta}}^{-1}$ ineffective, even incorrect. We substantiate this observation empirically in Section 4. Conversely, while $\nabla v^{\text{util}}$ in IP may not precisely align with the optimal direction for enhancing model performance, it remains a reliable indicator for distinguishing between detrimental and beneficial samples, even for non-convex models. Intuitively, if a sample gradient aligns with the gradient from the validation set, it suggests that incorporating this sample contributes to enhancing the model's utility. Similarly, the negative impact of a single sample on prediction can also be mitigated by considering its gradient.

**Extension**. Beyond measuring the sample influence on model utility, we extend IP to assess the sample influence on fairness and robustness by modifying the impact function $f$.

Specifically, we can instantiate the impact function $f$ by group fairness (Dwork et al., 2012), such as demographic parity (DP) to measure influence on fairness (Li & Liu, 2022). Consider a binary sensitive attribute defined as $g \in \{0, 1\}$ and let $\hat{y}$ denote the predicted class probabilities. The fairness metric DP is defined as: $f^{\text{DP-fair}}(\hat{\theta}, V) = \left| \mathbb{E}_V[\hat{y}|g=1] - \mathbb{E}_V[\hat{y}|g=0] \right|$. Within the above IP framework, we can calculate the training sample influence on fairness as follows:

$$\mathcal{I}_{\text{IP}}^{\text{DP-fair}}(-z_i) = \nabla v^{\text{fair}} \nabla_{\hat{\theta}} \ell(z_i; \hat{\theta}), \text{ and } \nabla v^{\text{fair}} = \nabla_{\hat{\theta}} f^{\text{DP-fair}}(\hat{\theta}, V)_{\hat{\theta}}^{\top}. \quad (3)$$

Similarly, we can also measure the sample influence on adversarial robustness within the IP framework. To achieve this, we follow Chhabra et al. (2024) and consider a white-box adversary (Megyeri et al., 2019) specific to linear models, which can be easily extended to other models and settings. To craft an adversarial sample, we take each sample $z_j = (x_j, y_j)$ in the validation set $V$ and only perturb $x_j' = x_j - \gamma \frac{\hat{\theta}^{\top} x_j + b}{\hat{\theta}^{\top} \hat{\theta}} \hat{\theta}$ and keep $y_j$ unchanged, where $\hat{\theta} \in \mathbb{R}^d$ are the linear model coefficients, $b \in \mathbb{R}$ is the intercept, and $\gamma > 1$ controls the amount of perturbation added. In this manner, we can obtain an adversarial validation set $V'$ which consists of $z_j' = (x_j', y_j)$ for each sample $z_j$ of $V$. Now, we can compute adversarial robustness influence for each training sample as follows:

$$\mathcal{I}_{\text{IP}}^{\text{robust}}(-z_i) = \nabla v^{\text{robust}} \nabla_{\hat{\theta}} \ell(z_i; \hat{\theta}), \text{ and } \nabla v^{\text{robust}} = \sum_{z_j' \in V'} \nabla_{\hat{\theta}} \ell(z_j'; \hat{\theta})^{\top}. \quad (4)$$

**Enhancement**. The simplicity of IP offers opportunities to enhance the generalization of influence functions. In convex cases, the model parameter $\hat{\theta}$ is both optimal and unique. However, in non-convex scenarios, the presence of local minima introduces instability and non-uniqueness into the solution. Typically, ensemble strategies are employed to bolster model generalization (Dietterich, 2000; Lakshminarayanan et al., 2017). Yet, while employing different models can enhance performance, it also escalates the costs associated with model training and complicates the calculation of influence functions. This arises from the variability in model parameters, necessitating multiple computations of the inverse of each individual Hessian matrix. The introduction of Hessian-free IP circumvents this issue, eliminating the need for costly calculations of Hessian matrices and their inverses. Drawing inspiration from dropout mechanisms, diverse models can be swiftly generated without necessitating model retraining. By computing sample gradients from various models, we propose IP Ensemble that amalgamates IP scores from distinct models. Experiments detailed in Section 5 illustrate the superior performance of the inner product ensemble over other influence function-based methods.

## 4 CORRECTNESS VERIFICATION ON SYNTHETIC DATA

Here we verify the correctness of our Inner Product (IP) as an influence score surrogate on two synthetic datasets with convex/non-convex models. First, we use Logistic Regression on a linear dataset to demonstrate the effectiveness of IP in a convex optimization scenario. Second, we use a

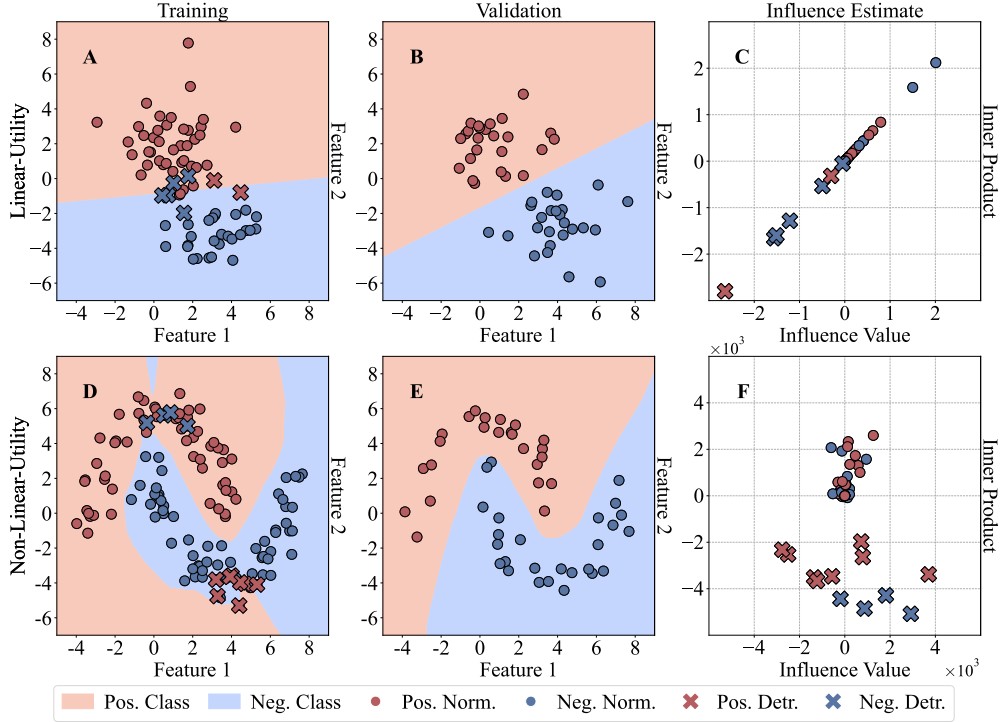

Figure 2: Illustrating our IP on two synthetic datasets and convex/non-convex models. **A**-**C** illustrate a 2D linearly separable synthetic dataset with a subset of detrimental samples bearing incorrect labels, trained using a Logistic Regression model, and **D**-**F** demonstrate the similar analysis on a non-linear synthetic half-moon dataset using a Multilayer Perceptron neural network. Specifically, **A** and **D** depict training sets with two classes, where detrimental samples are marked with × and regular samples with ○. **B** and **E** show test sets. **C** and **F** present influence scores and IP scores by Eqs. (1) and (2), respectively. In the linear case, there is a clear correlation between influence scores and inner product scores, the detrimental samples have both negative influence scores and IP scores. However, in the non-linear case, the influence scores of detrimental samples appear intermixed; fortunately, the detrimental samples are effectively isolated from inliers via IP.

Multi-Layer Perceptron on a half-moon dataset to validate the performance of IP in a non-convex optimization scenario.[1] In Figure 2, **A** and **B** illustrate the training and test sets of a linearly separable dataset, consisting of 150 and 100 samples, respectively. Notably, the training set contains 10 manually generated noisy samples with incorrect labels. Similarly, **D**-**F** illustrate a non-linear separable half-moons dataset, consisting of 200 training samples including 20 noisy samples with incorrect labels, and 100 test samples. **C** and **F** display the influence score and inner product score for each training sample, calculated according to Eqs. (1) and (2), respectively. We also verify the correctness of IP on model fairness and robustness in Appendix C.1.

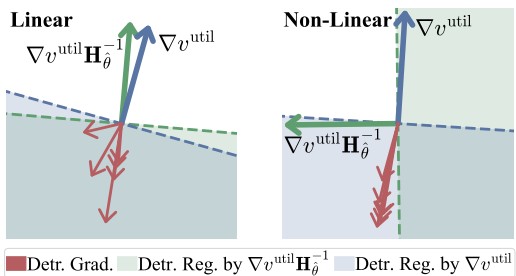

Figure 3: Directions of gradients of the validation set and detrimental samples in Figure 2. In the linear case, $\alpha$ is $0.66°$, and we draw a larger angle for better visualization. For the non-linear case, we use the dimension reduction (Bingham & Mannila, 2001) for visualization, and $\alpha$ is $94.36°$.

In the linear case depicted in Figure 2**C**, the inner product score serves as a reliable surrogate to distinguish detrimental samples from beneficial ones. It exhibits an almost perfect correlation and order-consistency with the influence score. Specifically, detrimental samples yield negative scores for both inner product and influence, while other samples typically show positive or nearly zero values. Figure 3 illustrates the relationship between directions among $\nabla v^{\text{util}}$, $\nabla v^{\text{util}} \mathbf{H}_{\hat{\theta}}^{-1}$, and the gradients

---

[1]The detailed characteristics of datasets and models used in this paper can be found in Appendix B.

Table 1: Accuracy results of influence function-based methods on the *CIFAR10N*, *CIFAR-100N* and *Animal-10N* datasets with 5% identified detrimental samples removed

| Methods / Datasets | CIFAR-10N-a | CIFAR-10N-r | CIFAR-10N-w | CIFAR-100N | Animal-10N | Avg. |
|---|---|---|---|---|---|---|
| Cross Entropy | 91.62 | 90.25 | 85.66 | 56.41 | 80.54 | 80.90 |
| LiSSA (Koh & Liang, 2017) | $92.13 \pm 0.29$ | $90.98 \pm 0.16$ | $85.97 \pm 0.47$ | $59.24 \pm 0.39$ | $81.93 \pm 0.14$ | 82.05 |
| TracIn (Pruthi et al., 2020) | $90.48 \pm 0.12$ | $88.09 \pm 0.24$ | $85.18 \pm 1.05$ | $56.47 \pm 1.87$ | $80.12 \pm 0.57$ | 80.07 |
| EKFAC (Grosse et al., 2023) | $91.76 \pm 0.23$ | $90.47 \pm 0.10$ | $83.25 \pm 0.38$ | $59.91 \pm 0.90$ | $80.89 \pm 0.54$ | 81.26 |
| DataInf (Kwon et al., 2023) | $91.88 \pm 0.39$ | $90.79 \pm 0.21$ | $86.22 \pm 0.13$ | $58.40 \pm 0.22$ | $81.60 \pm 0.23$ | 81.78 |
| Self-TracIn (Thakkar et al., 2023) | $92.03 \pm 0.09$ | $90.43 \pm 0.24$ | $86.00 \pm 0.18$ | $61.99 \pm 0.29$ | $81.82 \pm 0.34$ | 82.45 |
| Self-LiSSA (Bejan et al., 2023) | $91.91 \pm 0.17$ | $90.66 \pm 0.35$ | $85.73 \pm 0.41$ | $61.56 \pm 0.56$ | $81.23 \pm 0.24$ | 82.22 |
| TDA (Bae et al., 2024) | $91.95 \pm 0.19$ | $89.87 \pm 0.32$ | $84.02 \pm 0.41$ | $58.91 \pm 0.48$ | $80.57 \pm 0.25$ | 81.06 |
| GEX (Kim et al., 2024) | $91.81 \pm 0.27$ | $90.68 \pm 0.39$ | $85.64 \pm 0.20$ | $58.47 \pm 0.48$ | $80.78 \pm 0.58$ | 81.49 |
| IP (Ours) | $92.42 \pm 0.17$ | $90.82 \pm 0.08$ | $86.31 \pm 0.35$ | $60.59 \pm 0.20$ | $81.19 \pm 0.22$ | 82.27 |
| IP Ensemble (Ours) | $\mathbf{92.26} \pm 0.19$ | $\mathbf{91.28} \pm 0.29$ | $\mathbf{86.50} \pm 0.35$ | $\mathbf{62.25} \pm 0.54$ | $\mathbf{82.35} \pm 0.55$ | $\mathbf{82.93}$ |

of detrimental samples. It is worth noting that the angle between $\nabla v^{\text{util}} \mathbf{H}_{\hat{\theta}}^{-1}$ and $\nabla v^{\text{util}}$ is only $0.66°$, indicating that the identity matrix is a simple and effective surrogate of $\mathbf{H}_{\hat{\theta}}^{-1}$ in the linear case.

However, the limitations of influence scores become apparent in the context of non-convex models, as illustrated in Figure 2**F**. Here, the influence scores of detrimental samples are intermingled with those of normal ones, where $\nabla v^{\text{util}} \mathbf{H}_{\hat{\theta}}^{-1}$ is not the optimal direction due to the inaccuracies in approximating the Hessian matrix. Fortunately, the IP score effectively isolates detrimental samples from inliers. Even for this non-linear dataset, $\nabla v^{\text{util}}$ remains a useful indicator for discerning detrimental samples from beneficial ones, as all detrimental samples exhibit obtuse angles with $\nabla v^{\text{util}}$.

## 5 NOISY LABEL CORRECTION FOR VISION DATASETS

In this section, we demonstrate the effectiveness of our IP Ensemble in identifying detrimental samples on noisy vision datasets. Specifically, we choose three benchmark datasets *CIFAR-10N* (Wei et al., 2022), *CIFAR-100N* (Wei et al., 2022), and *Animal-10N* datasets (Shu et al., 2023) in the noisy label learning area. *CIFAR-10N* encompasses three distinct noise settings: aggregate, random, and worst, denoted as "-*a*," "-*r*," and "-*w*," respectively. "*a*" means that labels are derived via majority voting among three annotators, with ties being resolved randomly, "*r*" adopts the label provided by the first annotator, while "*w*" selects the label from the least reliable annotator.

For competitive methods, we choose the following influence function-based methods. TracIn (Pruthi et al., 2020) replaces the Hessian matrix with the identity matrix and considers checkpoints during the training process; LiSSA (Koh & Liang, 2017) and EKFAC (Grosse et al., 2023) employ implicit Hessian-vector products and Kronecker-Factored curvature to efficiently approximate the inverse of the Hessian matrix; DataInf (Kwon et al., 2023) swaps the order of the matrix multiple for obtaining a closed-form expression; Self-TracIn (Thakkar et al., 2023) and Self-LiSSA (Bejan et al., 2023) are the self-expression versions of TracIn and LiSSA, where $\nabla v^{\text{util}}$ is replaced with $\nabla_{\hat{\theta}} \ell(z_i; \hat{\theta})$ and only the last checkpoint, i.e., the converged model parameters, is used in Self-TracIn. GEX (Kim et al., 2024) utilizes ensemble methods based

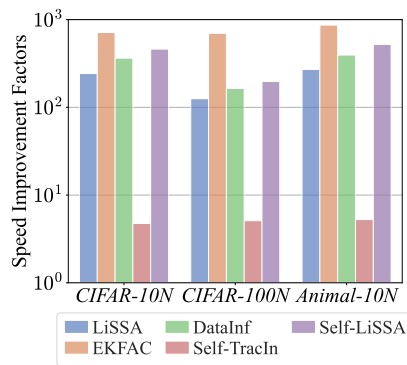

Figure 4: Speed improvement factors of IP over other baseline methods.

on checkpoints from extra stochastic gradient descent on the converged model. TDA (Bae et al., 2024) focuses checkpoints during the training process, ensembles the influence via EKFAC (Grosse et al., 2023). Our IP Ensemble method constitutes an ensemble version of IP with $\mathcal{U}(0, 0.01)$ dropout applied on model parameters and an ensemble size of 5.

To reduce randomness, we train a ResNet-34 network (He et al., 2016) once on each dataset to establish a baseline model. Subsequently, based on the same model, we employ the aforementioned influence function-based methods to identify 5% of detrimental samples. Following this, we conduct five retraining iterations of the ResNet-34 network, each time removing the identified detrimental samples from the training set. We report the average accuracy and standard deviation of the above

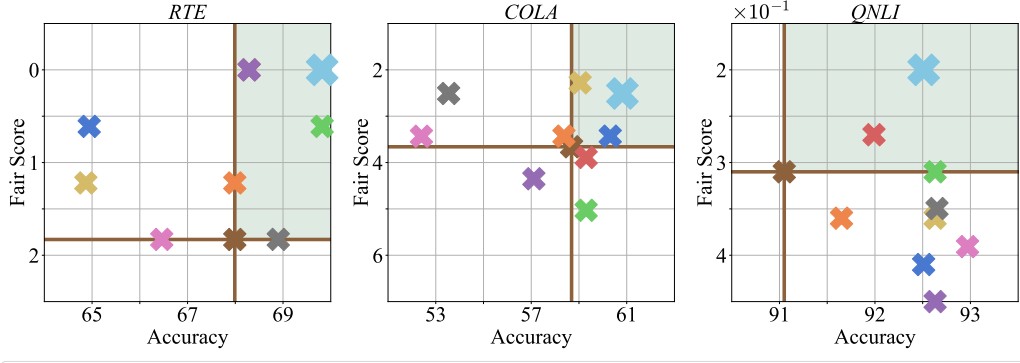

Figure 5: Accuracy and fair score of different influence function-based methods on fine-tuning *RTE*, *COLA*, and *QNLI* datasets. The X-axis denotes accuracy, and the Y-axis for fair score is inverted. The brown crossing denotes the performance of using all the samples for fine-tuning the RoBERTa model as the baseline model. Building upon this, we plot a horizontal and a vertical line in each figure and divide the space by fairness and utility results into four regions. The green area in the top right corner signifies a model that is both fairer and more accurate compared to the baseline model.

influence function-based methods across these five retrainings in Table 1. In general, these influence function-based methods are effective in identifying detrimental samples. Upon retraining the ResNet-34 model without these identified samples, nearly every result obtained by these methods outperforms the vanilla ResNet-34 trained on the entire dataset, except for EKFAC on *CIFAR-10N-w*. Notably, our IP Ensemble consistently outperforms other baseline methods across various noise conditions and datasets. Particularly noteworthy is the performance of our IP Ensemble in the most challenging scenario, *CIFAR-100N*, achieving the highest recorded accuracy of 62.25% on the test set, surpassing the vanilla cross-entropy accuracy of 56.41%. Moreover, TDA and TracIn only achieve moderate performance because they introduce multiple checkpoints throughout the training process. These approaches might dilute the effectiveness of identifying truly detrimental samples for the converged model, as the influence of each checkpoint may vary. Additionally, compared to IP, IP Ensemble delivers further performance gains, highlighting the benefits of the ensemble strategy for enhancing model generalization. Moreover, the average accuracy of the IP Ensemble on the test set reaches its peak at 82.93%, surpassing both the vanilla cross-entropy performance of 80.90% and the second-best accuracy of Self-TracIn at 82.45%. More experimental results and analysis on different percentages of removed samples, parameter analysis on different dropout rates, ensemble sizes, and base model architectures can be found in Appendix C.2- C.4, respectively.

In addition, we also present the running times of these influence function-based methods. Despite some baselines having linear time complexity, there is significant divergence in their real execution times. Given that our IP exhibits exceptional speed and similar execution times, we consider them as the baseline and compute the speed improvement factors over other baseline methods, as depicted in Figure 4. For ensemble methods including TracIN, TDA, GEX, and our IP Ensemble, parallel computation can be applied to accelerate the running time if enough resources are allowed; if calculated serially, the time grows linearly with the ensemble size. We do not report their running time in Figure 4. With the exception of Self-TracIn, our IP runs over 100 times faster than LiSSA, EKFAC, DataInf, and Self-LiSSA. Notably, on *Animal-10N*, IP is over 800 times faster than EKFAC. It is worth noting that Self-TracIn and Self-LiSSA are slower than their standard versions due to the fixed $\nabla v^{\text{util}}$ when calculating the influence of each sample. The simplicity and efficiency of IP make it a promising tool for analyzing sample influence on deep models. The time complexities and execution time of these methods can be found in Appendix C.5.

# 6    DATA CURATION TOWARDS FINE-TUNING OF FAIRER NLP MODELS

In this section, we further demonstrate the efficacy of our IP method in gauging the impact of individual samples on fairness within the realm of curating suitable data samples for fine-tuning language models. Beyond mere utility, fairness has emerged as an indispensable attribute for machine learning models to mitigate inadvertent discrimination.

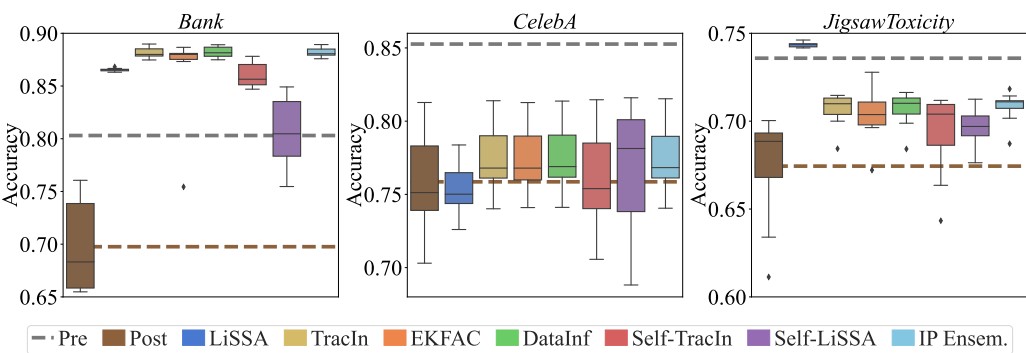

Figure 6: Performance variations across various influence function-based methods over 10 distinct attacks on *Bank*, *CelebA*, and *JigsawToxicity*. The dashed gray line presents the pre-attack performance, while the brown line denotes the average accuracy of post-attack over 10 runs.

In this experiment, we employ three datasets—*RTE*, *COLA*, and *QLIN*—from the GLUE repository (Wang et al., 2018) to fine-tune the widely-used language model RoBERTa (Liu et al., 2019). Our focus is on group fairness, necessitating that machine learning models treat samples within various predefined subgroups comparably. To assess fairness, we adopt the methodology outlined in (Qian et al., 2022), which involves perturbing the demographic information within each sample and scrutinizing whether the model yields identical predictions for the original sample $x$ and its corresponding perturbed counterpart $\tilde{x}$. This evaluation metric "fair score" is defined as $|\mathcal{C}(x) - \mathcal{C}(\tilde{x})|$ over all the samples in the test set, normalized by the size of the test set, where $\mathcal{C}(\cdot)$ is the model predictor. Note that the fair score is a negative metric, so smaller values are preferable. Utility and fairness serve as distinct perspectives for evaluating the performance of a model. Thus, in our fine-tuning experiments, we consider both utility and fairness. Employing the same influence function-based methods and our IP Ensemble as in the previous section, we conduct a comparative analysis. For each method, we calculate the influence on both utility and fairness within the fine-tuning set. We then identify and remove the 5% most detrimental samples in terms of utility and fairness before fine-tuning the RoBERTa model to optimize performance across both accuracy and fairness metrics.

The results of this experiment are presented in Figure 5, where the Y-axis for fairness is inverted. The brown crossing denotes the performance of using all the samples for model fine-tuning as the base model. Building upon this, we plot a horizontal and a vertical line in each figure and divide the space by fairness and utility results into four regions. The green area in the top right corner signifies a model that is both fairer and more accurate compared to the baseline model. Results within this green region can be considered Pareto improvements, enhancing both utility and fairness simultaneously. It is evident most results are located in the green area, indicating the existence of detrimental samples, and not all the samples are helpful to the model performance. This also implies that influence function methods are effective to identifying the detrimental samples, even for non-convex deep models. However, it is important to note that some competitive methods yield a trade-off or even Pareto deterioration. For instance, LiSSA demonstrates a better fair score but worse accuracy compared to the base model on *RTE*; conversely, it exhibits better accuracy but worse fairness on *QLIN*. EKFAC shows similar performance on *COLA* and *QLIN*. Self-LiSSA demonstrates Pareto deterioration on *COLA*. We hypothesize that the approximation of the inverse of the Hessian matrix may suffer from large errors, leading to heavily divergent influence estimations from the true values. Our IP Ensemble consistently achieves Pareto improvements across all three datasets, often yielding the best results compared to other influence function-based methods.

# 7 DEFENSE AGAINST ADAPTIVE EVASION ADVERSARIES

In this section, we demonstrate how the influence-based approach can effectively fortify defenses against an adaptive adversary (Tramer et al., 2020; Biggio et al., 2013) that performs evasion attacks on the learning model. In this scenario, the attacker randomly selects a subset of test samples to launch the evasion attack. We defend by proactively trimming the training set by a predetermined amount, although we lack specific knowledge about which samples are adversarial during inference.

Table 2: Defense performance of various influence function-based methods under the relabeling and reweighting strategies on *Bank*, *CelebA*, and *JigsawToxicity* datasets

| Defense Strategy | Relabeling | | | | Reweighting | | | |
|---|---|---|---|---|---|---|---|---|
| | *Bank* | *CelebA* | *JigsawToxicity* | Avg | *Bank* | *CelebA* | *JigsawToxicity* | Avg |
| Pre | 80.31 | 85.26 | 73.58 | 79.72 | 80.31 | 85.26 | 73.58 | 79.72 |
| Post | 70.79 ± 3.71 | 75.33 ± 3.04 | 66.18 ± 3.52 | 70.77 | 70.79 ± 3.71 | 75.33 ± 3.04 | 66.18 ± 3.52 | 70.77 |
| LiSSA (Koh & Liang, 2017) | 86.07 ± 0.39 | 68.68 ± 1.39 | **73.30** ± 2.96 | 76.02 | 78.69 ± 3.56 | 73.23 ± 3.70 | 70.13 ± 1.00 | 74.02 |
| EKFAC (Grosse et al., 2023) | 87.38 ± 0.89 | **77.36** ± 1.94 | 70.04 ± 0.96 | 78.26 | 83.50 ± 5.90 | **75.11** ± 1.73 | 70.05 ± 0.76 | 76.22 |
| DataInf (Kwon et al., 2023) | **87.46** ± 2.44 | **77.36** ± 1.85 | 70.39 ± 0.84 | 78.40 | 85.99 ± 2.49 | 74.74 ± 1.68 | 70.82 ± 0.35 | 77.18 |
| Self-TracIn (Thakkar et al., 2023) | 86.12 ± 1.10 | 75.58 ± 2.97 | 68.31 ± 2.73 | 76.67 | 85.22 ± 1.72 | 73.12 ± 3.71 | 71.37 ± 1.81 | 76.57 |
| Self-LiSSA (Bejan et al., 2023) | 78.97 ± 3.40 | 75.55 ± 2.99 | 69.18 ± 1.76 | 74.57 | 85.16 ± 1.74 | 73.17 ± 3.72 | **71.41** ± 1.86 | 76.58 |
| IP (Ours) | 87.45 ± 0.27 | 77.28 ± 1.87 | 70.43 ± 0.87 | 78.38 | 86.17 ± 2.04 | 75.10 ± 1.71 | 70.82 ± 0.36 | 77.36 |
| IP Ensemble (Ours) | 87.45 ± 0.32 | 77.30 ± 1.89 | 70.51 ± 1.00 | **78.42** | **86.44** ± 2.58 | 75.10 ± 1.73 | 70.67 ± 0.27 | **77.40** |

In this experiment, we utilize a Logistic Regression model with three datasets: *Bank* (Moro et al., 2014), *CelebA* (Liu et al., 2018), and *JigsawToxicity* (Noever, 2018). Following the protocol in Section 3, we consider a white-box adversary (Megyeri et al., 2019) to craft adversarial samples. For each sample $(x_j, y_j)$ in the validation set $\mathcal{V}$, we perturb it by changing the feature $x'_j = x_j - \gamma \frac{\hat{\theta}^\top x_j + b}{\hat{\theta}^\top \hat{\theta}} \hat{\theta}$ and keeping $y_j$ unchanged. The attacker perturbs between 5% and 25% of the test set samples at random. By quantifying the impact of samples on model robustness, we trim 5% detrimental samples in the training set through influence functions. The boxplot depicted in Figure 6 demonstrates the performance variations across various influence function-based methods over 10 distinct attacks. Since the Logistic Regression model is convex, TDA simplifies to EKFAC, and TracIn and GEX simplify to IP; therefore, we will omit their duplicate results in this figure and the following table. In general, influence function-based methods are highly effective against adaptive evasion attackers, showcasing superior resilience particularly post-attack, notably evident in scenarios involving *Bank* and *JigsawToxicity*. Among these influence function-based methods, our IP Ensemble demonstrates competitive efficacy with the best or the second best on these three datasets.

In addition to the trimming strategy, we continue to explore relabeling and reweighting strategies to tackle detrimental samples. The relabeling strategy (Kong et al., 2021) changes the identified detrimental samples from their original classes into another class, since the datasets we used here are binary classes, we directly flip their labels. The reweighting strategy (Thakkar et al., 2023) takes the influence score of each sample as the exponential weight with a softmax normalization, and then trains a model with weighted samples. We report the defense performance of various influence function-based methods and our IP Ensemble under the relabeling and reweighting strategies in Table 2. Under the relabeling strategy, DataInf achieves the best results on *Bank* and *CelebA* with 87.46% and 77.36% accuracy, respectively. However, LiSSA is not stable as other methods on *CelebA* with only 68.68%, which is even worse than the performance of post attack. Our IP ensemble method continues to achieve competitive performance, with accuracy scores of 87.45%, 77.30%, and 70.51% on three datasets with the second-best performance among all influence functions-based methods. For the reweighting strategy, the influence function-based methods are still effective on *Bank* and *JigsawToxicity*, which outperform the post-attack by a large margin, but achieve similar performance with post attack on *CelebA*. Notably, when considering the average accuracy of both the relabeling and reweighting strategies, our IP Ensemble method performs the best.

## 8 CONCLUSION

In this paper, we revisited, extended, and enhanced the TracIn method into the Inner Product (IP) formulation. By proving the exhibition of order-consistency in several scenarios between IP and influence function, we demonstrated substituting the inverse of the Hessian matrix with an identity matrix offers a practical and computationally efficient solution to estimating sample influence. Based on that, we extended our IP to measure the sample influence on fairness and robustness. Continually, we enhanced the generalization of IP by introducing an ensemble strategy. We verified the correctness of IP on synthetic datasets and extensive evaluations on noisy label detection, data curation for fair NLP model fine-tuning, and defense against adaptive adversarial attacks. Overall, our IP Ensemble highlighted the potential of simple, yet powerful, approximations in influence estimation.

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

# APPENDIX

## A  RATIONALITY OF INNER PRODUCT

In Figure 1, we visualize the direction of $\nabla v^{\text{util}} \mathbf{H}_{\hat{\theta}}^{-1}$ and $\nabla v^{\text{util}}$, with the influence scores roughly representing the angle between the sample gradient and $\nabla v^{\text{util}} \mathbf{H}_{\hat{\theta}}^{-1}$ or $\nabla v^{\text{util}}$, where $\mathbf{H}_{\hat{\theta}}^{-1}$ induces a rotation of $\alpha$ between these two directions. For a complete analysis, we further split Region II and IV into Sub-Regions as shown in Figure 7 assuming $\alpha < 45°$. Given gradients of two samples $z_i$ and $z_{i'}$, we analyze all possible scenarios as follows.

- **(I, I)**. For a case where both $z_i$ and $z_{i'}$ have gradients in Region I, $\mathcal{I}^{\text{util}}(-z_i) \gtreqqless \mathcal{I}^{\text{util}}(-z_{i'})$ implies $\mathcal{I}_{\text{IP}}^{\text{util}}(-z_{i'}) \gtreqqless \mathcal{I}_{\text{IP}}^{\text{util}}(-z_i)$, which is the opposite of order-consistency. While these samples are jointly recognized by $\mathcal{I}^{\text{util}}$ and $\mathcal{I}_{\text{IP}}^{\text{util}}$ as detrimental, there is little practical difference.
- **(I, II)**. For a case where $z_i$ has gradients in Region I and $z_{i'}$ has gradients in Region II, the order-consistency partially holds, depending on which Sub-Region the gradients of $z_{i'}$ is in. For Sub-Region II$^{\text{b}}$ and II$^{\text{c}}$, we have $\mathcal{I}^{\text{util}}(-z_i) > \mathcal{I}^{\text{util}}(-z_{i'})$ and $\mathcal{I}_{\text{IP}}^{\text{util}}(-z_i) > \mathcal{I}_{\text{IP}}^{\text{util}}(-z_{i'})$, and the order-consistency holds. For Sub-Region II$^{\text{a}}$, however, $\mathcal{I}^{\text{util}}(-z_i) \gtreqqless \mathcal{I}^{\text{util}}(-z_{i'})$ and $\mathcal{I}_{\text{IP}}^{\text{util}}(-z_i) > \mathcal{I}_{\text{IP}}^{\text{util}}(-z_{i'})$, thus the order-consistency partially holds. While samples in Sub-Region II$^{\text{a}}$ are jointly recognized by $\mathcal{I}^{\text{util}}$ and $\mathcal{I}_{\text{IP}}^{\text{util}}$ as detrimental, there is little practical difference.
- **(I, III)**. For a case where $z_i$ has gradients in Region I and $z_{i'}$ has gradients in Region III, $\mathcal{I}^{\text{util}}(-z_i) > \mathcal{I}^{\text{util}}(-z_{i'})$ and $\mathcal{I}_{\text{IP}}^{\text{util}}(-z_i) > \mathcal{I}_{\text{IP}}^{\text{util}}(-z_{i'})$, therefore the order-consistency holds.
- **(II, IV)**. For a case where $z_i$ has gradients in Region II and $z_{i'}$ has gradients in Region IV, the order-consistency partially holds.
- **(II, II)**. For a case where both $z_i$ and $z_{i'}$ have gradients in Region II, $\mathcal{I}^{\text{util}}(-z_i) \gtreqqless \mathcal{I}^{\text{util}}(-z_{i'})$ implies $\mathcal{I}_{\text{IP}}^{\text{util}}(-z_i) \gtreqqless \mathcal{I}_{\text{IP}}^{\text{util}}(-z_{i'})$, the order-consistency holds.

Based on the symmetry in samples ($z_i$ and $z_{i'}$) and regions (I and III, II and IV), we can get all order-consistencies in Table 3. The bold parts are discussed above, and the rest parts are from the symmetry. In all scenarios where order-consistency does not hold, samples are jointly recognized by both $\mathcal{I}^{\text{util}}$ and $\mathcal{I}_{\text{IP}}^{\text{util}}$ as either beneficial or detrimental, resulting in minimal practical difference.

Table 3: Order-consistency in different scenarios

| Gradient of $z_i$ | Gradient of $z_{i'}$ | Order-Consistency |
| --- | --- | --- |
| Region **I** | Region **I** | Not Hold |
| Region **I** | Region **II**, IV | Partially Hold |
| Region **I** | Region **III** | Hold |
| Region **II** | Region I, III, **IV** | Partially Hold |
| Region **II** | Region **II** | Hold |
| Region III | Region III | Not Hold |
| Region III | Region II, IV | Partially Hold |
| Region III | Region I | Hold |
| Region IV | Region I, II, III | Partially Hold |
| Region IV | Region IV | Hold |

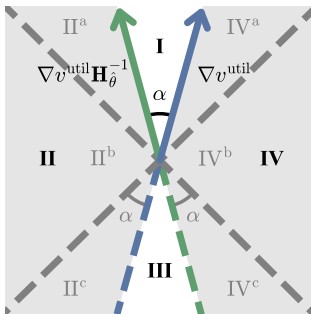

Figure 7: Visualization of the directions $\nabla v^{\text{util}} \mathbf{H}_{\hat{\theta}}^{-1}$ and $\nabla v^{\text{util}}$.

## B  DETAILED INFORMATION ON DATASETS AND MODEL TRAINING

We describe dataset details, model training, and other information used in the main paper, below.

### B.1  DATASETS

We cover our generated synthetic datasets in Section 4, vision datasets in Section 5, text datasets in Section 6, and datasets for robustness in Section 7.

#### B.1.1  SYNTHETIC DATASETS

We generate two synthetic datasets to validate the correctness of our IP method in measuring the convex and non-convex models' utility, fairness, and robustness. Specifically, we generate a linear

dataset using the scikit-learn (Pedregosa et al., 2011) library's `make_blobs` function, which consists of 150 training samples and 100 test samples. The second dataset is the non-linear *half moons* dataset so that we can train a Multi-Layer Perception network with two hidden layers with ReLU activations. The training set has 250 samples and the test set has 100 samples, and the dataset is generated using the scikit-learn library's `make_moons` function. Here we manually flip the labels of 20 samples (10 from each class) to add noise to the data.

### B.1.2 VISION DATASETS

Both the *CIFAR-10N* and *CIFAR-100N* datasets (Oliver et al., 2018) consist of the same input images that make up the *CIFAR-10* (10 classes) and *CIFAR-100* (100 classes) datasets (Krizhevsky et al., 2009), respectively. Each input is a 32x32 RGB image with a dimension of (3,32,32). However, for *CIFAR-10N* and *CIFAR-100N*, the labels are noisy, as they contain real-world human annotation errors collected using 3 annotators on Amazon Mechanical Turk. As these datasets are based on human-annotated noise, they model noisy real-world datasets more realistically, compared to synthetic data alternatives. The training set for both datasets contains 50,000 image-label pairs, and the test set contains 10,000 image-label pairs that are free from noise. For *CIFAR-10N* we utilize three noise settings for experiments in the paper– (1) *Worst* is the dataset version with the highest noise rate (40.21%) as the worst possible annotation label for the image is chosen, (2) *Aggregate* is the least noisy dataset (9.03%) as labels are chosen via majority voting amongst the annotations, and (3) *Random* has intermediate noise (17.23%) and consists of picking one of the annotators' labels. We use the first annotator for the random labels. For *CIFAR-100N* there is only a single noisy setting due to the large number of labeling classes, and the overall noise rate is 40.20%.

### B.1.3 NLP DATASETS

For the three GLUE datasets—*RTE*, *CoLA*, and *QNLI*(Wang et al., 2018), we perturbe the validation set as follows. The *Recognizing Textual Entailment* (*RTE*) dataset consists of sentence pairs labeled as entailment or not entailment. It is derived from a series of annual textual entailment challenges. The dataset includes 2,490 training examples and 277 validation examples. Since the test set does not have labels, we split the validation set into two parts, with one half used as the validation set for computing influence, and the other half used as the test set. Subsequently, we generated perturbed datasets for both the validation set and the test set using a seq2seq model (Qian et al., 2022). The *Corpus of Linguistic Acceptability* (*CoLA*) dataset consists of sentences labeled as grammatically acceptable or unacceptable. This dataset is derived from publications in linguistics and includes 8,551 training examples and 1,043 validation examples. As with *RTE*, the test set for CoLA does not have labels, so we split the validation set into two parts, using one half as the validation set for computing influence, and the other half as the test set. Perturbed datasets were generated using a seq2seq model (Qian et al., 2022) for both the validation set and the test set. The *Question Natural Language Inference* (*QNLI*) dataset is a large-scale corpus for question answering, consisting of question-sentence pairs from the Stanford Question Answering dataset. The task is to determine whether the context sentence contains the answer to the question. It includes 104,743 training examples and 5,463 validation examples. Similar to the other datasets, the test set does not have labels, so we split the validation set into two parts, using one half as the validation set for computing influence, and the other half as the test set. Perturbed datasets were also generated using a seq2seq model (Qian et al., 2022) for both the validation set and the test set. These datasets represent diverse natural language understanding tasks, and using them helps evaluate the fairness and utility of the fine-tuned RoBERTa model.

### B.1.4 DATASETS FOR DEFENCING ATTACKS

We utilize three datasets: *Bank*, *CelebA*, and *JigsawToxicity*, to evaluate the defense against adaptive evasion adversaries. The *Bank* dataset (Moro et al., 2014) consists of features extracted from direct marketing campaigns of a Portuguese banking institution. The goal is to predict whether a client will subscribe to a term deposit. The dataset includes 18,292 training examples, 6,098 validation examples, and 6,098 test examples. The feature dimension of the dataset is 51. Both the validation and test sets have labels. The *CelebA* dataset (Liu et al., 2018) is a large-scale face attributes dataset with more than 200,000 celebrity images, each annotated with 40 binary attributes. For this experiment, we focus on a subset of these images. The dataset is split into 62,497 training examples, 20,833 validation examples, and 20,833 test examples. The feature dimension of the dataset is 39. Both

the validation and test sets have labels. The *JigsawToxicity* dataset (Noever, 2018) contains a large number of comments from Wikipedia labeled for toxicity. The goal is to predict the toxicity level of a given comment. The dataset includes 18,000 training examples, 6,000 validation examples, and 6,000 test examples. The feature dimension of the dataset is 385. These datasets cover a range of tasks from marketing prediction and face attribute recognition to toxicity detection, providing a robust evaluation of the model's defense mechanisms against adaptive adversaries.

## B.2 MODELS AND METHODS

We now describe the models and the methods used in our experiments throughout the main paper. First, we describe the ResNet-34 (He et al., 2016) architecture used as the base model for the noisy vision datasets, then the RoBERTa (Liu et al., 2019) NLP transformer model. We also describe implementation details and parameter values for the label correction baselines used in Sections 7 and the influence-based baselines used throughout the paper.

### B.2.1 MLP

In our experiments as described in Section 4, we utilized a three-layer Multi-Layer Perceptron (MLP) classifier. This MLP is structured with three fully connected layers, each followed by a ReLU activation function. Specifically, the input layer maps the input features to 32 neurons, the hidden layer maintains this dimensionality, and the output layer is a single neuron producing the final prediction. The final layer's output is passed through a sigmoid activation function to yield a probability score.

### B.2.2 RESNET-34

The ResNet-34 model in Section 5 was proposed in (He et al., 2016) and is a 34-layer convolutional neural network pretrained on the ImageNet-1K dataset at resolution $224 \times 224$. The pretrained model block is fine-tuned on the *CIFAR-10N/CIFAR-100N* training set experiments with default parameters–minibatch size (128), optimizer (SGD), initial learning rate (0.1), momentum (0.9), weight decay (0.0005), and number of epochs (100), for all experiments.

### B.2.3 ROBERTA

We use the Roberta-base model from Huggingface[2] in Section 6. The learning rate is 0.00001 and the batch sizes of *RTE*, *CoLA*, and *QNLI* are 64, 16, and 32. The model is fine-tuned over 10 epochs. The loss function used is a negative log-likelihood as the datasets are all for binary classification.

### B.2.4 LOGISTIC REGRESSION

The Logistic Regression model in Section 7 is implemented using the `sklearn` library. The model uses L2 regularization with and a maximum iteration limit of 2,048 (Chhabra et al., 2024).

### B.2.5 INFLUENCE-BASED BASELINES

In our experiments, we utilize the following influence function-based methods as baselines: TracIn (Pruthi et al., 2020) replaces the Hessian matrix with the identity matrix; LiSSA (Koh & Liang, 2017) and EKFAC (Grosse et al., 2023) use implicit Hessian-vector products and Kronecker-Factored curvature to efficiently approximate the inverse of the Hessian matrix; DataInf (Kwon et al., 2023) swaps the order of matrix multiplication to obtain a closed-form expression; Self-TracIn (Thakkar et al., 2023) and Self-LiSSA (Bejan et al., 2023) are self-expression versions of TracIn and LiSSA, where $\nabla v^{\text{util}}$ is replaced with $\nabla_{\hat{\theta}} \ell(z_i; \hat{\theta})$. GEX (Kim et al., 2024) utilizes ensemble methods based on checkpoints from extra stochastic gradient descent on the converged model. TDA (Bae et al., 2024) focuses checkpoints during the training process, ensembles the influence via EKFAC (Grosse et al., 2023). Our IP closely resembles TracIn and GEX, with the added application of dropout with $\mathcal{U}(0, 0.01)$ on the model parameters. TracIn, GEX, IP, and IP Ensemble are all Hessian-free and they

---

[2]`https://huggingface.co/docs/transformers/model_doc/roberta`

are ensemble versions, except for IP. Here, we present their calculations to highlight the differences between them as follows:

$$\mathcal{I}_{\text{IP}}^{\text{util}}(-z_i, \hat{\theta}) = \sum\nolimits_{z_j \in V} \nabla_{\hat{\theta}} \ell(z_j; \hat{\theta})^{\top} \nabla_{\hat{\theta}} \ell(z_i; \hat{\theta}),$$

$$\mathcal{I}_{\text{TracIn}}^{\text{util}}(-z_i, \Theta_{\text{TracIn}}) = \frac{1}{T} \sum\nolimits_{\hat{\theta}_t \in \Theta_{\text{TracIn}}} \mathcal{I}_{\text{IP}}^{\text{util}}(-z_i, \hat{\theta}_t),$$

$$\mathcal{I}_{\text{GEX}}^{\text{util}}(-z_i, \Theta_{\text{GEX}}) = \frac{1}{T} \sum\nolimits_{\hat{\theta}_t \in \Theta_{\text{GEX}}} \mathcal{I}_{\text{IP}}^{\text{util}}(-z_i, \hat{\theta}_t),$$

$$\mathcal{I}_{\text{IP Ensemble}}^{\text{util}}(-z_i, \Theta_{\text{IP Ensemble}}) = \frac{1}{T} \sum\nolimits_{\hat{\theta}_t \in \Theta_{\text{IP Ensemble}}} \mathcal{I}_{\text{IP}}^{\text{util}}(-z_i, \hat{\theta}_t),$$

(5)

where $\hat{\theta}$ in IP is the converged model parameters; $\Theta_{\text{TracIn}}$ denotes the set of model parameters from the saved checkpoints during model training; $\Theta_{\text{GEX}}$ is obtained by training the converged model for several extra iterations; IP Ensemble gets $\Theta_{\text{IP Ensemble}}$ by applying dropout mechanisms on the model; $T$ is the ensemble size. The advantages of IP Ensemble are that it does not require recording checkpoints compared to TracIn, and it does not need extra model training processes compared to GEX. We argue that the early checkpoints might not be helpful to analyze the sample influence and the extra iterations on the converged model make little effects.

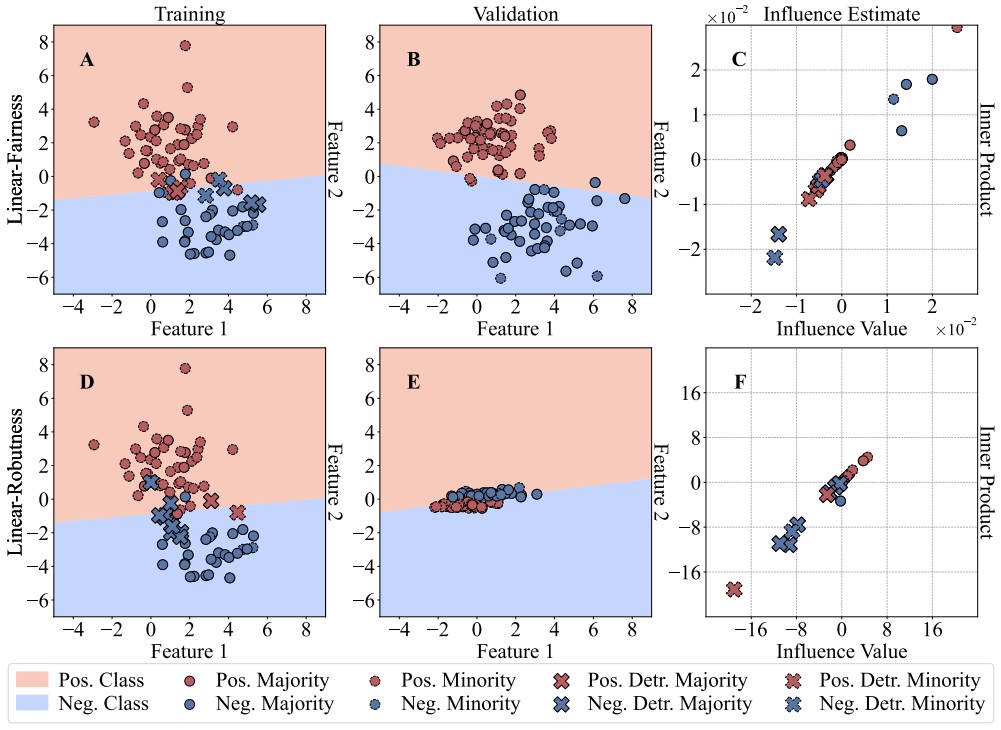

Figure 8: Illustrating our inner product approach on measuring fairness and robustness. **A** and **D** illustrate the same 2D linearly separable synthetic dataset in Figure 2 trained using a Logistic Regression model for binary classification, where the solid and dashed point boundaries denote the majority and minority subgroups. **B** and **E** represent the validation set. **C** and **F** show the estimated influence on fairness and robustness by the traditional influence function and our IP method.

## C    ADDITIONAL EXPERIMENTAL RESULTS AND ANALYSIS

### C.1    IP FOR MEASURING FAIRNESS AND ROBUSTNESS

Similar to Figure 2, we conduct the experiments based on Eqs. (3) and (4) to demonstrate the effectiveness of our IP in measuring the sample influence on fairness and robustness. Figure 8 shows

the relationship between IP and the traditional influence function, indicating IP is a good surrogate of the traditional influence function in measuring fairness and robustness as well.

## C.2 PERFORMANCE OF IP ENSEMBLE WITH DIFFERENT RATES OF REMOVED SAMPLES

In our paper, we explore different rates of removed samples on *CIFAR-10N-a*. Table 4 shows the performance of IP Ensemble with different rates of removed samples. Except for not removing, there is no significant difference in the remaining results.

Table 4: Performance of IP Ensemble with different rates of removed samples on *CIFAR-10N-a*

| Method | Ensemble Size | Remove Rate | ACC |
|---|---|---|---|
| Cross Entropy | 5 | 0 | 91.62 |
| IP Ensemble (Ours) | 5 | 0.025 | $92.29 \pm 0.16$ |
| IP Ensemble (Ours) | 5 | 0.050 | $92.26 \pm 0.19$ |
| IP Ensemble (Ours) | 5 | 0.075 | $92.50 \pm 0.13$ |
| IP Ensemble (Ours) | 5 | 0.100 | $92.37 \pm 0.15$ |

## C.3 PARAMETER ANALYSIS ON DROPOUT RATE AND ENSEMBLE SIZE

Table 5 shows the performance of different dropout rates and ensemble sizes of IP Ensemble. As can be observed, our IP Ensemble is not sensitive to dropout rate, and its performance increases with a large ensemble size, demonstrating the effectiveness of the ensemble strategy. It is also worth noting that although our IP Ensemble runs fast, it might take a longer time to calculate the sample gradient.

Table 5: Performance of different dropout rate and ensemble size of IP Ensemble on *CIFAR-10N-a*, *CIFAR-10N-r*, *CIFAR-10N-w*, and *CIFAR-100N*

| Ensemble Size | Dropout Rate | *CIFAR-10N-a* | *CIFAR-10N-r* | *CIFAR-10N-w* | *CIFAR-100N* |
|---|---|---|---|---|---|
| 5 | 0.01 | $92.26 \pm 0.19$ | $91.28 \pm 0.29$ | $86.50 \pm 0.35$ | $62.25 \pm 0.54$ |
| 5 | 0.1 | $92.26 \pm 0.19$ | $91.28 \pm 0.29$ | $86.50 \pm 0.35$ | $62.25 \pm 0.54$ |
| 5 | 0.5 | $92.26 \pm 0.19$ | $91.28 \pm 0.29$ | $86.50 \pm 0.35$ | $62.25 \pm 0.54$ |
| 1 | 0.01 | $92.42 \pm 0.17$ | $90.82 \pm 0.08$ | $86.31 \pm 0.35$ | $60.59 \pm 0.20$ |
| 5 | 0.01 | $92.26 \pm 0.19$ | $91.28 \pm 0.29$ | $86.50 \pm 0.35$ | $62.25 \pm 0.54$ |
| 10 | 0.01 | $92.58 \pm 0.04$ | $91.32 \pm 0.29$ | $86.89 \pm 0.37$ | $61.82 \pm 0.61$ |
| 15 | 0.01 | $92.27 \pm 0.09$ | $91.27 \pm 0.28$ | $86.47 \pm 0.41$ | $61.59 \pm 0.34$ |
| 20 | 0.01 | $92.41 \pm 0.15$ | $91.33 \pm 0.26$ | $86.65 \pm 0.16$ | $62.25 \pm 0.11$ |

## C.4 PERFORMANCE ON ViT AND MLP-MIXER

To verify the effectiveness of our IP ensemble across different network architectures, we conducte experiments using ViT (Dosovitskiy, 2020) and MLP-Mixer (Tolstikhin et al., 2021) on various vision datasets in Table 6. We use a batch size of 512 for all experiments in this section. The models are trained on *CIFAR-10N* and *CIFAR-100N* datasets for 100 epochs, and on *Animal-10N* for 400 epochs. The learning rate was set to $1 \times 10^{-4}$ for ViT and $1 \times 10^{-3}$ for MLP-Mixer.

Our IP Ensemble consistently demonstrates superior performance compared to the vanilla models and other baseline methods on different base models. Specifically, when trained on ViT, IP Ensemble achieves an average accuracy of 70.06%, outperforming the vanilla ViT's average accuracy of 68.87%. This improvement is observed across all datasets, with notable gains on *CIFAR-100N* (51.01% versus 48.53%) and *Animal-10N* (77.20% versus 76.20%). Additionally, compared to baseline methods, our IP Ensemble achieves the best performance with ViT or MLP-Mixer based models, indicating the effectiveness of our methods on different base models.

Table 6: Accuracy results of influence function-based methods with ViT and MLP-mixer as based models on *CIFAR10N-a*, *CIFAR-100N*, and *Animal-10N* with 5% detrimental samples removed.

| Methods / Datasets | CIFAR-10N-a | CIFAR-100N | Animal-10N | Avg. |
|---|---|---|---|---|
| ViT | 81.87 | 48.53 | 76.20 | 68.87 |
| LiSSA (Koh & Liang, 2017) | $81.83 \pm 0.19$ | $48.47 \pm 0.24$ | $76.90 \pm 0.35$ | 69.07 |
| TracIn (Pruthi et al., 2020) | $81.59 \pm 0.21$ | $48.18 \pm 0.28$ | $76.21 \pm 0.58$ | 68.66 |
| EKFAC (Grosse et al., 2023) | $80.79 \pm 0.56$ | $48.90 \pm 0.30$ | $76.85 \pm 0.30$ | 68.85 |
| DataInf (Kwon et al., 2023) | $81.68 \pm 0.21$ | $49.00 \pm 0.22$ | $76.70 \pm 0.27$ | 69.13 |
| Self-TracIn (Thakkar et al., 2023) | $81.88 \pm 0.12$ | $49.25 \pm 0.21$ | $76.95 \pm 0.25$ | 69.36 |
| Self-LiSSA (Bejan et al., 2023) | $82.89 \pm 0.12$ | $49.30 \pm 0.18$ | $76.80 \pm 0.28$ | 69.66 |
| TDA (Bae et al., 2024) | $81.89 \pm 0.28$ | $49.05 \pm 0.54$ | $76.44 \pm 0.32$ | 69.12 |
| GEX (Kim et al., 2024) | $\mathbf{82.99} \pm 0.24$ | $49.78 \pm 0.20$ | $76.88 \pm 0.29$ | 69.88 |
| IP (Ours) | $81.93 \pm 0.22$ | $49.12 \pm 0.20$ | $76.80 \pm 0.22$ | 69.28 |
| IP Ensemble (Ours) | $81.96 \pm 0.13$ | $\mathbf{51.01} \pm 0.17$ | $\mathbf{77.20} \pm 0.24$ | **70.06** |
| MLP-Mixer | 74.42 | 35.63 | 72.69 | 60.91 |
| LiSSA (Koh & Liang, 2017) | $75.35 \pm 0.47$ | $36.50 \pm 0.21$ | $72.89 \pm 0.28$ | 61.58 |
| TracIn (Pruthi et al., 2020) | $74.43 \pm 0.28$ | $35.21 \pm 0.35$ | $72.71 \pm 0.43$ | 60.78 |
| EKFAC (Grosse et al., 2023) | $75.49 \pm 0.28$ | $36.34 \pm 0.24$ | $73.12 \pm 0.19$ | 61.65 |
| DataInf (Kwon et al., 2023) | $75.10 \pm 0.45$ | $35.78 \pm 0.29$ | $73.17 \pm 0.21$ | 61.35 |
| Self-TracIn (Thakkar et al., 2023) | $75.88 \pm 0.21$ | $36.72 \pm 0.19$ | $73.81 \pm 0.54$ | 62.14 |
| Self-LiSSA (Bejan et al., 2023) | $75.41 \pm 0.38$ | $36.05 \pm 0.27$ | $72.56 \pm 0.34$ | 61.34 |
| TDA (Bae et al., 2024) | $74.89 \pm 0.31$ | $36.84 \pm 0.38$ | $71.88 \pm 0.43$ | 61.20 |
| GEX (Kim et al., 2024) | $75.57 \pm 0.27$ | $36.85 \pm 0.25$ | $\mathbf{73.89} \pm 0.43$ | 62.10 |
| IP (Ours) | $75.04 \pm 0.23$ | $36.15 \pm 0.18$ | $73.47 \pm 0.20$ | 61.55 |
| IP Ensemble (Ours) | $\mathbf{75.77} \pm 0.19$ | $\mathbf{37.12} \pm 0.18$ | $73.80 \pm 0.11$ | **62.23** |

## C.5 TIME COMPLEXITY AND EXECUTION TIME OF VARIOUS INFLUENCE FUNCTION-BASED METHODS ON VISION DATASETS

Table 7 shows the time complexity of various influence function-based methods. Except for the vanilla calculation, all other methods have linear time complexity in terms of the sample size. However, they have large divergence in real execution time.

We omit the specific timing details for the sample gradient, which are readily available during the base model's training phase. Besides, utilizing vmap in Pytorch, we can efficiently compute gradients in parallel. For example, in Section 5, it only takes 61 seconds and 4 seconds to calculate $\nabla v^{\text{util}}$ and $\nabla_{\hat{\theta}} \ell(z_i; \hat{\theta})$ on *CIFAR-10N* dataset, respectively.

Table 7: Computational complexity of influence-function-based methods ($n$ is #training samples and $p$ is #model parameters, $k$ is #checkpoints or #ensemble size with $k=1$ here. "-" denotes no runs.)

| Method | Type | Time Complexity | CIFAR-10N | CIFAR-100N | Animal-10N |
|---|---|---|---|---|---|
| Exact by Eq. (1) | Hessian-based | $\mathcal{O}(np^3)$ | - | - | - |
| LiSSA (Koh & Liang, 2017) | Hessian-based | $\mathcal{O}(np)$ | 7.67 | 34.59 | 7.46 |
| TracIn (Pruthi et al., 2020) | Hessian-free | $\mathcal{O}(npk)$ | 0.03 | 0.28 | 0.03 |
| EKFAC (Grosse et al., 2023) | Hessian-based | $\mathcal{O}(np^2)$ | 22.54 | 192.58 | 23.89 |
| DataInf (Kwon et al., 2023) | Hessian-based | $\mathcal{O}(np)$ | 11.50 | 45.29 | 10.89 |
| Self-TracIn (Thakkar et al., 2023) | Self-influence | $\mathcal{O}(npk)$ | 0.15 | 1.41 | 0.15 |
| Self-LiSSA (Bejan et al., 2023) | Self-influence | $\mathcal{O}(np)$ | 14.57 | 54.39 | 14.40 |
| GEX (Kim et al., 2024) | Hessian-free | $\mathcal{O}(npk)$ | 0.03 | 0.28 | 0.03 |
| TDA (Bae et al., 2024) | Hessian-based | $\mathcal{O}(np^2k)$ | 23.98 | 193.79 | 24.77 |
| **IP Ensemble (Ours)** | Hessian-free | $\mathcal{O}(npk)$ | 0.03 | 0.28 | 0.03 |

## D BROADER IMPACT AND LIMITATIONS

Our work aims to address issues that currently hinder the applicability of influence estimation in deep learning models. By enabling influence estimation for deep models, practitioners can assess whether training samples are beneficial or detrimental to performance. As we demonstrate through extensive experiments across multiple problem settings, our proposed outlier gradient

analysis approach outperforms existing baselines and can augment model performance by trimming detrimental samples. As a result, our work paves the way for significant positive societal impact, especially with the increased adoption of larger and deeper neural networks such as LLMs.

However, despite our research demonstrating superior performance and potential benefits, there are several limitations to consider. First, our method assumes a level of model and data homogeneity that might not hold in highly heterogeneous datasets or more complex model architectures. Furthermore, there is the consideration of robustness and fairness across different domains and types of data; our current evaluations, while extensive, may not cover all possible scenarios or edge cases, potentially limiting the generalizability of our findings. Finally, ethical considerations around data trimming and sample selection must be carefully managed to avoid unintended biases or negative impacts on model performance for underrepresented groups. As influence estimation becomes more integrated into model training and evaluation pipelines, ongoing research and monitoring will be essential to ensure that these techniques are applied responsibly and equitably.

Due to the extensive computational demands of our experiments (see details in Appendix E), which require multiple retraining cycles to evaluate model stability and robustness, it becomes impractical for us to conduct comparative experiments on very large datasets, e.g., ImageNet. The limited computational resources at our disposal make it challenging to perform such large-scale retraining efforts, especially given the significant training time and GPU requirements. As a result, we focus our evaluation on middle-size yet representative benchmarks that allow for more feasible experimentation while maintaining the rigor of our comparative analyses.

## E  CODE AND REPRODUCIBILITY

We provide our code, instructions, and implementation in an open-source repository: https://anonymous.4open.science/r/IP_ensemble-3E48/README.md. The experiments were conducted on a Linux (Ubuntu 20.04.6 LTS) server using NVIDIA GeForce RTX 4090 GPUs with 24GB VRAM running CUDA version 12.3 and driver version 545.23.08.

