# OpenReview forum: "Revisit, Extend, and Enhance Hessian-Free Influence Functions"
_ICLR.cc/2025/Conference — Submitted to ICLR 2025_

### Official Review · Reviewer_kr1G · 2024-10-16

**Soundness:** 1
**Presentation:** 3
**Contribution:** 1
**Rating:** 3
**Confidence:** 5

**Summary:**

The authors introduce Inner Product (IP), a metric that simplifies the classic influence function by dropping the inverse Hessian term to assess the influence of training samples. They validate IP through sanity checks on synthetic data and demonstrate its superior performance across various applications, including noisy label detection, fairness enhancement, and defense against evasion attacks.

**Strengths:**

- The sanity checks on synthetic data yield some interesting findings, e.g., for non-linear models, IP can separate corrupted samples from normal samples whereas influence function cannot
- The experiments are comprehensive and cover quite a few applications (nevertheless, I think the paper is mostly following Chhabra et al. 2024)

**Weaknesses:**

If IP is indeed as useful as the authors were suggesting in this paper, this would be a breakthrough for the field of influence function (and more broadly, data attribution), since the (inverse) Hessian term is indeed the main bottleneck of applying influence functions to large-scale models. Unfortunately, after reading the paper, I am not convinced that IP is a principled and useful metric.

- The authors confused the relationship between influence function and TracIn, and incorrectly viewed TracIn as an approximation of influence function. Influence function is a first-order approximation of leave-one-out, which measures influence w.r.t. changes to the training set. On the other hand, TracIn is a first-order approximation of the change of loss, which measures the change in model parameters as a function of time. They are fundamentally different metrics since they have completely definition of "influence". The authors seemed to suggest that IP can be derived from both influence function (by replacing the inverse Hessian with identity), and TracIn (by only considering the last iteration). Regardless of whether such simplifications make sense or not, this makes IP hard to interpret --- the basic question is, what notion of "influence" is IP adopting, and what ground truth is it approximating?

- The analysis at the beginning of P4 is completely wrong, as there is no way that IP can preserve order even in specific regions. For instance, in Figure 1, one can easily construct two gradient vectors in Region II as follows: the first vector is orthogonal to the blue vector and has small norm, the second vector forms an acute angle with the blue vector and has large norm. By adjusting the norm, we can reverse the order. The main issue is: removing the inverse Hessian must lose information, and there is no analysis on why such information could potentially be useless.

- The ensemble procedure is unclear and confusing. The procedure is not formally explained; I assume that the dropout is only applied to the final model checkpoint, since the authors mentioned "diverse models can be swiftly generated without necessitating model retraining". What is the rationale behind this approach? In the literature, people do multiple retraining and ensemble to reduce the inherent noise in training and influence function estimation. It is unclear to me that applying multiple dropout to the final model checkpoint would enjoy the same benefit.

- Finally, regarding the experimental results, the improvement compared to existing approaches is very limited. In table 1, all methods look similar to me after taking into account the variance.

There are also some minor points:

- I think the authors should read [1] carefully to develop a better understanding of the literature

- Some algorithms which have similar design (e.g., ensemble) are missing, for instance, TRAK [2]

- Some claims lack evidence, for instance, "We believe that sample influence should be assessed using a fixed model; simply summing sample influences at different stages fails to capture the dynamics of model optimization" --- it seems to me that the essence of TracIn is exactly being dynamic

References

[1] Hammoudeh, Zayd, and Daniel Lowd. "Training data influence analysis and estimation: A survey." Machine Learning 113.5 (2024): 2351-2403.

[2] Park, Sung Min, et al. "Trak: Attributing model behavior at scale." arXiv preprint arXiv:2303.14186 (2023).

**Questions:**

The paper clearly cannot be accepted in its current form, but I think a few results on synthetic data in Section 4 look interesting. The authors could explore the following question: under what circumstance (on data and/or model), the directions of $H^{-1}v$ and $v$ are close.

---

> ### Author Response · Authors · 2024-11-24
> **Response(1/2) to Reviewer kr1G**
>
> We would like to thank the reviewer for the deep analysis of our work, and the time, effort, and consideration. Below, we answer the questions raised by the reviewer:
>
> **Influence in IP**
> Thank you for the insightful interpretation on influence function and TracIn. We fully agree! From the formulation, our IP is similar to influence function and TracIn. Here we can provide another intuition of our method. If the training samples are similar to validation samples, we expect the model trained with such samples can also perform well on the validation set. Note that the similarity should be calculated not in the original feature space, where this space is orthogonal to the learning model. Therefore, the cosine similarity in the gradient space measures how the training sample is similar to the validation set in the point of the model, i.e., the gradient space. This approach circumvents the instability of Hessian inverse computations while providing an intuitive and computationally efficient alternative.
>
> **Order consistency**
> Thank you for the careful thought. In this paper, we do not claim the order consistency always holds. Moreover, in our synthetic datasets, we empirically demonstrated the linear case might have the order consistency, but the non-linear case not.
>
> We agree with Reviewer HFTU that order consistency might not be a good interpretation. We will remove this part.
>
> **Remove the inverse of Hessian**
> We agree with Reviewer KR1G that the inverse of Hessian matrix contains information. However, computing the inverse of the Hessian matrix in non-convex scenarios is notoriously challenging and unstable, particularly when the Hessian matrix contains near-zero or negative eigenvalues, leading to numerical instability. This instability can introduce additional noise, undermining the reliability of influence computation. The cost of approximating the inverse of Hessian matrix is even higher than directly removing it. From our empirical evaluation, our IP does not suffer from too much performance sacrifice but has simple formulation. We will revise the manuscript to articulate these points more clearly.
>
> **Ensemble**
> Your understanding is correct. Our ensemble method generates diverse models by applying dropout to the final model checkpoint, avoiding the traditional approach of selecting different checkpoints from the training process. The rationale behind this design is as follows:
>
> Traditional methods often rely on selecting multiple checkpoints during training to generate diverse models. However, these checkpoints are typically derived from non-converged model states, which may contain significant optimization noise, thereby affecting the accuracy of influence function estimation. In contrast, our method is based on the final converged model and leverages the stochasticity of dropout to simulate uncertainty in model parameters, creating diverse models efficiently. This approach mitigates the noise introduced by non-converged states while avoiding the high computational cost of retraining. Experimental results demonstrate that our method achieves robust influence estimation and significantly improves final model performance (see Figure 3 and Figure 4). In the revised manuscript, we will provide a more formal definition of the ensemble procedure and include additional experimental analyses to illustrate the applicability and potential limitations of this method.
>
> **Improvement very limited**
> Our major contribution lies in that IP is a simplified version of TracIn and works well in empirical verification. We agree with Reviewer KR1G that the inverse of Hessian matrix contains information, but approximating it might bring higher cost than removing it. Our IP without this term can achieve better and comparable performance, where we consider this simplicity and effectiveness to be a key strength of our approach. If we can achieve similar performance with a simpler tool, there would be no reason to add unnecessary components to the existing complex tool.
>
> **Survey**
> This survey paper is a key reference for our work. In fact, it first caught our attention two years ago when the arXiv version was made available. We are pleased to hear that the paper has now been accepted by Machine Learning. We would like to kindly inquire about which section Reviewer KR1G suggests we revisit or re-learn. Thank you.

---

> > ### Author Response · Authors · 2024-11-24
> > **Response(2/2) to Reviewer kr1G**
> >
> > **TRAK**
> > We report the performance of TRAK on CIFAR-10 and CIFAR-100 in the table below. In general, TRAK employs the random project for acceleration and suffers from the performance drop compared with the unaccelerated version.
> >
> > | Methods / Datasets     | CIFAR-10N-a          | CIFAR-10N-r          | CIFAR-10N-w          | CIFAR-100N           |
> > |-------------------------|----------------------|----------------------|----------------------|----------------------|
> > | Cross Entropy          | 91.62               | 90.25               | 85.66               | 56.41               |
> > | TRAK                   | 90.86 ± 0.15        | 89.42 ± 0.22        | 83.73 ± 0.50        | 59.12 ± 1.02        |
> > | IP (Ours)              | 92.42 ± 0.17        | 90.82 ± 0.08        | 86.31 ± 0.35        | 60.59 ± 0.20        |
> > | IP Ensemble (Ours)     | **92.26 ± 0.19**    | **91.28 ± 0.29**    | **86.50 ± 0.35**    | **62.25 ± 0.54**    |
> >
> > **Unclear claims**
> > Apologies for any confusion. We would like to clarify that the influence of samples varies during the model optimization process. Some samples may be beneficial in the early stages but become detrimental later on. Specifically, when model parameters change significantly, the influence of a sample in the early stages does not accurately reflect its true impact on the converged model. Simply summing the sample influences at different checkpoints, as done in TracIn, can lead to a compromise that results in inferior performance. We have verified this through our experiments and will make this point clearer in the paper.

---

> > ### Comment · Reviewer_kr1G · 2024-11-24
> >
> > > However, computing the inverse of the Hessian matrix in non-convex scenarios is notoriously challenging and unstable, particularly when the Hessian matrix contains near-zero or negative eigenvalues, leading to numerical instability. This instability can introduce additional noise, undermining the reliability of influence computation. The cost of approximating the inverse of Hessian matrix is even higher than directly removing it.
> >
> > When making this claim, the authors seem to overlook the vast body of literature dedicated to accelerating the computation of the inverse Hessian matrix (e.g., [1, 2]). While I recognize that this remains a significant challenge in influence functions, the assertion that "the cost of approximating the inverse of the Hessian matrix is even higher than directly removing it" lacks sufficient supporting evidence. At the very least, the authors should provide a discussion on why existing acceleration methods are deemed unsatisfactory.
> >
> > > Our major contribution lies in that IP is a simplified version of TracIn and works well in empirical verification.
> >
> > I assume the authors would agree that their work falls under the broader topic of "training data attribution" (TDA). In my opinion, while demonstrating the utility of TDA algorithms in downstream tasks (e.g., mislabeled data detection, improving fairness/robustness, etc.) is valuable, focusing **solely** on these applications without explicitly specifying the influence notion the metric adopts and the counterfactual it predicts is extremely problematic. As anticipated, the authors did not conduct any counterfactual evaluation, likely because it remains unclear what the "ground truth" (e.g., LOO, LDS) should be in this context. I would not be surprised if IP performs poorly on such metrics.
> >
> > The mislabeled data detection task is a strong example supporting this viewpoint (this is also pointed out by y3uC). From Table 1, it seems that the direct loss-based approach (using cross-entropy) is the most effective for this task, achieving good performance with virtually no computational cost. My own experience also supports this observation: in a simple binary classification task with some corrupted labels, this loss-based approach even outperforms influence functions in identifying mislabeled samples. However, it is evident that this approach is not well-suited for training data attribution, as it ignores other critical dimensions of a sample's contribution. This highlights the concern that downstream applications can be "hacked", meaning improvements in task-specific metrics do not necessarily correspond to better TDA algorithms.
> >
> > Finally, if the authors genuinely believe that IP is a valuable metric deserving broader consideration by the community, I strongly recommend conducting large-scale experiments on models like CLIP and T5. If the experimental setup is unclear, referring to the TRAK paper may provide useful guidance. Without explicitly defining influence and counterfactuals, such comprehensive experiments are the only way to convincingly demonstrate that IP has potential utility.
> >
> > [1] Kwon, Yongchan, et al. "Datainf: Efficiently estimating data influence in lora-tuned llms and diffusion models." arXiv preprint arXiv:2310.00902 (2023).
> >
> > [2] Grosse, Roger, et al. "Studying large language model generalization with influence functions." arXiv preprint arXiv:2308.03296 (2023).

---

> > > ### Author Response · Authors · 2024-11-29
> > >
> > > **Overlook the vast body of literature**
> > > Thank you for your valuable feedback! Regarding the computation of the inverse Hessian matrix, we have demonstrated the effects of directly removing it in Figure 2 C and F:
> > >
> > > - In Figure 2 C, we show that in convex optimization scenarios, our method maintains a high correlation with classical influence function results even without the inverse Hessian, while significantly reducing computational costs.
> > > - In Figure 2 F, we illustrate that in non-convex settings, our method (IP) outperforms influence functions that rely on the inverse Hessian. This indicates that computing the inverse Hessian in non-convex optimization is highly challenging due to numerical instability caused by near-zero or negative eigenvalues. In such cases, directly removing the inverse Hessian improves the reliability of the results.
> > >
> > > We compared [1,2] in our experiments. Existing works (e.g., [1, 2]) on approximating the inverse Hessian often incorporate regularization terms to address these numerical instabilities. However, such regularization alters the intrinsic properties of the inverse Hessian, and due to the lack of ground truth, the quality of these approximations remains unverifiable. By directly removing the inverse Hessian, our method avoids these issues and demonstrates superior efficiency and robustness in practice.
> > >
> > > We will expand on the limitations of existing acceleration methods in the revision and provide additional discussion to clarify the advantages of our approach.
> > >
> > > ---
> > >
> > > **LOO or LDS**
> > > We would like to share our personal opinion on why LOO or LDS is not suitable for data attribution evaluation. Although the purposes of LOO or LDS are reasonable, they are impractical. For LOO, the actual influence by model retraining is not accurate [1], if considering the randomness in the optimization. LDS is calculated by multiple runs with different subsets with a certain size, which is sensitive to the selected subsets.
> > >
> > > Since data attribution has no ground truth for each sample, we prioritize utilizing the downstream task for evaluation.
> > >
> > > [1] Influence Functions in Deep Learning Are Fragile.
> > >
> > > ---
> > >
> > > **Mislabeled Data Detection Task**
> > > Respectfully, it seems inappropriate to conclude that mislabeled data detection is a not well-suited task if the existing influence function methods do not perform well on this task. We agree with the reviewer that some influence function methods by removing their identified samples are worse than the vanilla one with the complete training set. On the contrary, Table 1 is good supporting evidence, showing our proposed IP Ensemble outperforms the vanilla one and other influence function methods. This just indicates the limitations of the existing methods on the inverse of Hessian matrix approximation and the effectiveness of our method.
> > >
> > > ---
> > >
> > > **Large-scale experiments on models like CLIP and T5**
> > > In Section 6, we provided the data selection for LLM model fine-tuning. Could you please clarify the advantage of experiments on CLIP and T5 over the current experiment we conducted? Humbly, we do not see any substantial strengths doing this. But in another ongoing project, we indeed employed CLIP and T5 datasets for large model pre-training by extending and further simplifying the IP approach. Even the current IP struggles for large model pre-training, let alone other methods with the approximation of the inverse of the Hessian matrix.

---

### Official Review · Reviewer_Hftu · 2024-10-28

**Soundness:** 2
**Presentation:** 4
**Contribution:** 2
**Rating:** 3
**Confidence:** 3

**Summary:**

This paper proposes a variation of TracIN called IP, where the ensembling over checkpoints is abandoned in favor of the converged model.
As this particular variation has appeared several times as baseline in ablation experiments in prior works (e.g. in both TracIN and the original influence function paper), the authors attempt to revisit this formulation with new theoretical and experimental insights.
Some intuition is provided for why this simple setup might work through an argument for similarity to the inverse hessian influence function. Further extensions of IP in the application of fairness, and a more powerful ensemble version of IP, are also presented. Experimentally, the proposed method is competitive with prior art in the benchmarked tasks in terms of downstream performance.

+++++++++++++++++++++++++++++++++++++++++++++++++++++

I have read the author's rebuttal. While I agree with some of the points made by the authors, I'm opting to maintain my original review score as the contribution of the paper and the experimental evidence for impact is still unclear

**Strengths:**

1. The proposed method is particularly simple, easy to implement and efficient to compute.
2. The experimental evaluations presented in the paper are fairly thorough and rigorous, with appropriate repeat experiments to establish confidence intervals.
3. The extension of influence estimation to algorithmic fairness metrics is interesting.

**Weaknesses:**

1. The order consistency argument for why IP is a good approximation to inverse hessian influence is quite weak. In figure 1, data points with vectors in regions I and III would not satisfy order consistency. The authors argue that since IP and IF both rate such points as beneficial/detrimental, the order doesn’t matter. This is clearly not true, as many applications of influence estimation involve determining set membership at the extreme ends of the influence spectrum (e.g. removing x% detrimental examples based on the most negative influence values or curating x% most beneficial examples for fine tuning based on the highest influence values). Changes to ordering in these regions are in fact more significant than changes to ordering within regions II and IV, which would be assigned neutral influence scores.
2. Furthermore, the conditions under which the angle alpha is small has not been discussed.
3. While the benchmarks on downstream utility are quite extensive, other quantitative metrics relating to similarity of predictions, such as correlation to the (inverse hessian) influence function, leave-one-out training, and precision-recall curves of label perturbation detection, are missing.

**Questions:**

What does U(0, 0.01) on line 318 mean in the context of model parameter drop out? Are model parameters set to zero with a chance of 1%, with or without replacement?

---

> ### Author Response · Authors · 2024-11-24
> **Response to Reviewer Hftu**
>
> We would like to thank the reviewer for the deep analysis of our work, and the time, effort, and consideration. Below, we answer the questions raised by the reviewer:
>
> **Order consistency**
> Thank you for the careful thought. In this paper, we do not claim the order consistency always holds. Moreover, in our synthetic datasets, we empirically demonstrated the linear case might have the order consistency, but the non-linear case not.
>
> We agree with Reviewer HFTU that order consistency might not be a good interpretation. We will remove this part. But here we can provide another intuition of our method. If the training samples are similar to validation samples, we expect the model trained with such samples can also perform well on the validation set. Note that the similarity should be calculated not in the original feature space, where this space is orthogonal to the learning model. Therefore, the cosine similarity in the gradient space measures how the training sample is similar to the validation set in the point of the model, i.e., the gradient space. This approach circumvents the instability of Hessian inverse computations while providing an intuitive and computationally efficient alternative.
>
> **Conditions under which the angle alpha is small**
> See the above response.
>
> **Correlation to the inverse Hessian**
> In non-convex scenarios, the inverse of the Hessian matrix is notoriously difficult and unstable to compute. The indefinite nature of the Hessian makes its inverse highly sensitive to small perturbations. It is impossible to calculate the approximation error or correlation to the inverse of Hessian matrix for complex data. Instead, we employ the downstream tasks (i.e., model performance without identified detrimental samples) to compare different algorithms.
>
> **On Leave-one-out training**
> Thank you for your thoughtful suggestion. However, we respectfully and frankly believe that leave-one-out training may not be an effective evaluation tool. From a computational perspective, leave-one-out training is prohibitively expensive, especially for large datasets and complex models, as it requires multiple retraining cycles. Additionally, in non-linear cases, particularly with deep learning, removing a single training sample may not have a significant impact, especially when considering the inherent randomness in the training process. In other words, we conducted experiments using ResNet on CIFAR-10, comparing performance with the complete training set versus with one sample removed. After running these experiments multiple times, we found that the average performance between the two settings did not show statistical significance.
>
> **Label perturbation detection**
> Our research theme focuses on goal-oriented data-centric learning, aiming to improve model performance by distinguishing between beneficial and detrimental samples. This is fundamentally different from noisy label detection, as these concepts address separate issues. Note that a sample with a correct label might still be a detrimental sample. Thus, the requested metric does not align with our research focus.
>
> **U(0, 0.01)**
> We will formally define the IP Ensemble process below and will include a clearer explanation in future revisions. This process involves the following steps:
>
> 1. **Generating Diverse Models:**
>    For each ensemble model, a dropout rate $p_k$ is randomly sampled from a uniform distribution $U(0, 0.01)$. The dropout is then applied to the model parameters $\theta$ to generate diverse models:
>    $$\theta^{(k)} = \text{Dropout}(p_k, \theta), \quad k = 1, 2, \dots, K.$$
>
> 2. **Computing IP Scores for Each Model:**
>    For each diverse model $\theta^{(k)}$, the IP score of a training sample $z_i$ is computed as:
>    $$I_{\text{IP}}^{(k)}(-z_i) = \nabla_{v_{\text{util}}}^{(k)} \cdot \nabla_{\theta^{(k)}} \ell(z_i; \theta^{(k)}),$$
>    where
>    $$\nabla_{v_{\text{util}}}^{(k)} = \sum_{z_j \in V} \nabla_{\theta^{(k)}} \ell(z_j; \theta^{(k)})^\top$$
>    is the aggregated gradient of the validation set $V$ based on the $k$-th model.
>
> 3. **Ensembling the IP Scores:**
>    The final IP Ensemble score for $z_i$ is obtained by averaging the IP scores from all $K$ models:
>    $$I_{\text{IP-Ensemble}}(-z_i) = \frac{1}{K} \sum_{k=1}^K I_{\text{IP}}^{(k)}(-z_i).$$

---

> ### Author Response · Authors · 2024-11-29
>
> Thank you for your acknowledgment on reading our response.

---

### Official Review · Reviewer_DTqB · 2024-10-30

**Soundness:** 1
**Presentation:** 2
**Contribution:** 1
**Rating:** 3
**Confidence:** 4

**Summary:**

The paper proposes an approach for measuring influence of samples in trained models by defining it as the inner product between the gradient of a specific sample and the sum of gradients from all other samples. By substituting this gradient sum with a fairness measure, the method claims to capture the influence of a sample on the model's fairness. When this gradient sum is replaced with gradients of adversarially perturbed samples, the method is intended to reflect the impact on robustness. Additionally, the paper suggests enhancing the proposed measurement through an ensemble of models. Experiments were conducted for applications such as detecting noisy labels, curating fairness-sensitive samples, and improving resilience against adversarial attacks.

**Strengths:**

The paper includes a lot of experiments and provides statistical ranges for the reported results.

**Weaknesses:**

However, the paper lacks a clear problem statement and positioning of the method within existing approaches. The formula provided to describe the method is the TracIn formula, with the only difference being the reduction of calculations to the final trained model. This raises several questions: (i) the explanation of the benefits of this simplification is vague and could be questioned, (ii) the intuition behind this approximation rests on a comparison with a method involving the Hessian’s inverse and assumes cosine similarity between vectors without a clear justification, and (iii) there is no explanation of what this formula intuitively means in contexts of fairness or adversarial robustness. Furthermore, the paper does not formally define the process for generating ensembles to accumulate the influence scores.

A precise definition of the influence function and an explanation of the role and significance of the inverse Hessian in this context would improve the method’s description.

Section 4, which highlights differences between this approach and an influence method that incorporates the Hessian, presents results on synthetic data and a simple multi-layer perceptron. It shows that Hessian based method fails due to approximation errors. However, it does not clearly demonstrate or discuss the approximation errors, and it is unclear why a setup without approximation errors cannot be tested on more complex, non-linearly separable data. Figure 3 lacks sufficient explanation, diminishing clarity.

In subsequent experiments where the ensemble inner product method shows improvements, only the original method appears beneficial in terms of time complexity. Moreover, as the ensemble size increases, it closely approximates TracIn with multiple measurements, making the specific contributions of this paper unclear.

Section 6 presents improvements in fairness, yet the benchmarks primarily focus on accuracy, limiting the validity of the comparison. Finally, the approach appears to offer limited benefits in adversarial defense.

Minor Issues: The related work section includes descriptions of studies that do not effectively contextualize the current paper’s contributions.

**Questions:**

What is the precise role and interpretation of the influence function? How does the inverse Hessian factor into the formula?

---

> ### Author Response · Authors · 2024-11-24
> **Response (1/2) to Reviewer DTqB**
>
> We would like to thank the reviewer for the deep analysis of our work, and the time, effort, and consideration. Below, we answer the questions raised by the reviewer:
>
> **Benefits of this simplification**
> Our simplified approach calculates influence values based solely on the final converged model. The rationale is that only a converged model has stable parameters that can accurately reflect the true influence of samples on the model's predictions. As highlighted in [1], during non-converged states, parameter fluctuations can lead to unstable influence values, thereby reducing the reliability of the analysis. Our experimental results further support this conclusion. The IP Ensemble method consistently outperforms TracIn across nearly all scenarios presented in our paper.
>
> **Intuition**
> We have validated the effectiveness of removing the Hessian inverse not only on synthetic datasets but also on real-world datasets. These experiments demonstrate that influence value evaluation remains accurate and reliable even without the computation of the Hessian inverse. Here we can provide another intuition of our method. If the training samples are similar to validation samples, we expect the model trained with such samples can also perform well on the validation set. Note that the similarity should be calculated not in the original feature space, where this space is orthogonal to the learning model. Therefore, after the removal of the Hessian inverse, the cosine similarity measures how the training sample is similar to the validation set in the point of the model, i.e., the gradient space. This approach circumvents the instability of Hessian inverse computations while providing an intuitive and computationally efficient alternative.
>
> **Fairness or adversarial robustness**
> We follow [2] to use influence function to measure the sample influence towards fairness and robustness. For instance, under different loss functions, influence functions can quantify the contribution of samples to model decisions, thereby demonstrating their role in fairness or adversarial robustness tasks. To make this point clear, we will address this more thoroughly in future revisions.
>
> **Ensemble process**
> In the current version, we provide a simplified description of the ensemble process. However, the process itself is relatively direct, involving the aggregation of influence scores from different models to improve robustness and stability. We will formally define this process below and will include a clearer explanation in future revisions. This process involves the following steps:
>
> 1. **Generating Diverse Models:**
>    For each ensemble model, a dropout rate $p_k$ is randomly sampled from a uniform distribution $U(0, 0.01)$. The dropout is then applied to the model parameters $\theta$ to generate diverse models:
>    $$\theta^{(k)} = \text{Dropout}(p_k, \theta), \quad k = 1, 2, \dots, K.$$
>
> 2. **Computing IP Scores for Each Model:**
>    For each diverse model $\theta^{(k)}$, the IP score of a training sample $z_i$ is computed as:
>    $$I_{\text{IP}}^{(k)}(-z_i) = \nabla_{v_{\text{util}}}^{(k)} \cdot \nabla_{\theta^{(k)}} \ell(z_i; \theta^{(k)}),$$
>    where
>    $$\nabla_{v_{\text{util}}}^{(k)} = \sum_{z_j \in V} \nabla_{\theta^{(k)}} \ell(z_j; \theta^{(k)})^\top$$
>    is the aggregated gradient of the validation set $V$ based on the $k$-th model.
>
> 3. **Ensembling the IP Scores:**
>    The final IP Ensemble score for $z_i$ is obtained by averaging the IP scores from all $K$ models:
>    $$I_{\text{IP-Ensemble}}(-z_i) = \frac{1}{K} \sum_{k=1}^K I_{\text{IP}}^{(k)}(-z_i).$$
>
> This approach not only avoids the computational cost of retraining but also improves the robustness of influence estimation by leveraging model diversity introduced through dropout.
>
> **Approximation errors**
> Computing the inverse of the Hessian matrix in non-convex scenarios is notoriously challenging and unstable. The indefinite nature of the Hessian makes its inverse highly sensitive to small perturbations. Note that it is likely possible that the inverse of Hessian matrix does not exist without extra regularization to ensure its validity. Even with such regularization, approximation errors in the Hessian inverse can significantly impact the influence function results.
>
> In Section 4, we employ a synthetic dataset to validate the effectiveness of removing the Hessian inverse in a controlled experimental environment. For real-world data and models, the Hessian matrix is of high dimension, which brings in the difficulty in computation and visualization. Therefore, it is impossible to calculate the approximation error of the inverse of Hessian matrix for complex data. Instead, we employ the downstream tasks (i.e., model performance without identified detrimental samples) to compare different algorithms.

---

> > ### Author Response · Authors · 2024-11-24
> > **Response (2/2) to Reviewer DTqB**
> >
> > **Figure 3**
> > Figure 3 aims to demonstrate the ineffectiveness of the inaccurate inverse of Hessian matrix in the non-linear case. The red arrows denote the gradient direction of detrimental samples with incorrect labels. An effective identification of detrimental samples requires \( \nabla v^{\textrm{util}} \mathbf{H}^{-1}_{\Hat{\theta}} \) and the sample gradient forms an obtuse angle. However, the inaccurate inverse of Hessian matrix forms an acute angle.
> >
> > **Contributions**
> > Our major contribution lies in that IP is a simplified version of TracIn and works well in empirical verification. Here we consider this simplicity to be a key strength of our approach.
> >
> > Compared to TracIn, our method demonstrates superior performance, particularly in tasks involving fairness and adversarial robustness. While increasing the ensemble size might make our method resemble TracIn's multiple measurements in form, our approach is fundamentally different. By relying solely on the final converged model, our method avoids TracIn's reliance on multiple intermediate optimization states, significantly reducing computational and storage costs. More importantly, the ensemble strategy in our method further enhances model performance, showing strong adaptability to complex tasks in experiments. We highlight the major difference between our method and TracIn as follows:
> >
> > - **Simplification and Reformulation:** We simplify TracIn’s multi-checkpoint computation by relying solely on the final converged model, reformulating it into an inner product (IP) framework. Furthermore, we delve into the rationale behind this simplification and provide both theoretical and practical insights into why it performs well.
> > - **Framework Extension:** We extend the IP framework beyond traditional applications of measuring sample influence on model utility to encompass tasks related to fairness and robustness. These extensions go beyond the original scope of TracIn.
> > - **Ensemble Innovation:** Building on this, we propose IP Ensemble, a novel approach leveraging dropout mechanisms to generate diverse models. By aggregating IP scores from these models, IP Ensemble significantly enhances generalization capabilities, a feature absent in the original TracIn framework.
> > - **Experimental Validation:** We validate the IP framework not only on synthetic datasets but also through extensive real-world experiments. These include noisy label correction for vision data, data curation for fairer NLP model fine-tuning, and defense strategies against adversarial attacks.
> >
> > **Related work**
> > Thanks for this point. We will add some descriptions in the related work to highlight the position of our method compared with the literature.
> >
> > **Precise role and interpretation of the influence function**
> > The influence function [1] evaluates the impact of a single training sample on the model's performance on validation data with the linear assumption. The inverse Hessian in the influence function incorporates second-order information from the loss function, accounting for the curvature of the parameter space. This term enables the influence function to accurately capture the sample's true impact in the linear case while considering parameter dependencies. Intuitively, the inverse Hessian re-weights the training sample gradient to improve reliability and interpretability. However, when it comes to the non-linear case, the inverse of Hessian matrix likely does not exist and some approximations/regularizations/assumptions are required to pursue influence scores.
> >
> > References:
> > [1] Pang Wei Koh and Percy Liang. Understanding black-box predictions via influence functions. In *International Conference on Machine Learning*, 2017.
> > [2] Anshuman Chhabra, Peizhao Li, Prasant Mohapatra, and Hongfu Liu. What data benefits my classifier? Enhancing model performance and interpretability through influence-based data selection. In *International Conference on Learning Representations*, 2024.

---

> > > ### Comment · Reviewer_DTqB · 2024-11-26
> > >
> > > I thank the authors for the reply.
> > >
> > > I will wait for a discussion with other reviewers to make a decision.

---

> > > > ### Author Response · Authors · 2024-11-29
> > > >
> > > > Thank you for your detailed feedback and for providing us with the opportunity to address your concerns! We look forward to the discussion among the reviewers and are happy to make further improvements based on any subsequent suggestions.

---

### Official Review · Reviewer_y3uC · 2024-11-03

**Soundness:** 2
**Presentation:** 3
**Contribution:** 1
**Rating:** 3
**Confidence:** 5

**Summary:**

This paper proposed a simplified method (IP) for training data attribution. The method equipped a hessian-free method to avoid the large computational cost on the Hessian matrix and its inverse. The paper also proposes some tricks (e.g., ensemble) and more applications (e.g., change the target function to robustness/fairness). Some empirical studies are also proposed to show the efficiency improvement and performance.

**Strengths:**

- The paper is well written, and the reader can understand the main idea of the paper quickly in a short time.
- Efficiency has become a very important topic for TDA(training data attribution).

**Weaknesses:**

- There is a large gap between the contribution claimed in this paper and actual literature. The major problem lies in the first and second contribution bullet point in section 1 (line 59 - line 62)
  - The Inner Product (IP) proposed by this paper (as a simplified version of TracIN) has long been proposed [1] and used in a large number of papers[2].
  - Replacement of the loss gradient to some other metrics to fairness and robustness is also something tried for influence function or related methods[3].
- The experiment is not fully fair and may intentionally make other baseline methods to have lower results. Furthermore, the improvement is kind of trivial.
  - The dropout trick can also be used on other methods as well, what are their performance?
  - Some related gradient-based SOTA methods (e.g., TRAK) are ignored. Furthermore, random projection has been a very widely used method for efficiency, but it is also omitted in the paper.
 - The manually added noisy data for Figure 2 (D) seems to be cherry-picked. The negative performance of H^-1 seems to be very unlikely.


Overall the contribution over-claiming is kind of the major problem for this paper, and that’s the main reason why I give a 3 point assessment.


[1] Charpiat, G., Girard, N., Felardos, L., & Tarabalka, Y. (2019). Input similarity from the neural network perspective. Advances in Neural Information Processing Systems, 32.

[2] Kwon, Y., Wu, E., Wu, K., & Zou, J. (2023). Datainf: Efficiently estimating data influence in lora-tuned llms and diffusion models. arXiv preprint arXiv:2310.00902.

[3] Richardson, B., Sattigeri, P., Wei, D., Ramamurthy, K. N., Varshney, K., Dhurandhar, A., & Gilbert, J. E. (2023, June). Add-Remove-or-Relabel: Practitioner-Friendly Bias Mitigation via Influential Fairness. In Proceedings of the 2023 ACM Conference on Fairness, Accountability, and Transparency (pp. 736-752).

**Questions:**

- Could you elaborate on this “We believe that sample influence should be assessed using a fixed model; simply summing sample influences at different stages fails to capture the dynamics of model optimization.” more? It seems hard to understand.
- Noisy label detection and other downstream tasks that do not have a very large difference between each method. Maybe some counterfactual prediction evaluation metrics (e.g., LDS) are more suitable?

---

> ### Author Response · Authors · 2024-11-24
> **Response to Reviewer y3uC**
>
> We would like to thank the reviewer for the deep analysis of our work, and the time, effort, and consideration. Below, we answer the questions raised by the reviewer:
>
>
> **Gap between the contribution claimed in this paper and actual literature**. We believe Reviewer Y3UC understood our method. Clearly, we do not claim that we propose TracIn; instead, our major contribution lies in that IP is a simplified version of TracIn and works well in empirical verification. We suspect that Reviewer Y3UC may view our method as too simple. However, we consider this simplicity to be a key strength of our approach. If we can achieve similar performance with a simpler tool, there would be no reason to add unnecessary components to the existing complex tool.
>
> Thanks for providing these references. For [1], it primarily focuses on measuring the similarity between samples, rather than directly addressing data valuation. We will cite it in our paper. For [2], we have already cited the relevant works, discussed their advantages and limitations in our paper (Line 573), and compare it in the experiments. For [3], we understand there are several existing studies that employ influence functions to estimate sample impact on fairness and robustness. Here we follow [4] to extend our IP method for fairness and robustness as well, where we briefly mentioned this in the contribution part.
>
> **Unfair Experiments**. We solicit Reviewer Y3UC for more evidence, which helps for our informative response, rather than just opinion. Here we follow the setting of [4] in a fair comparison. Again, our contribution lies in the simplicity, and our IP achieves better or comparable performance.
>
> **Dropout trick**. We agree that the dropout trick can also be applied to other baseline methods. We would like to note that it is impractical to apply an ensemble strategy for complicated methods.
>
> **Figure 2 (D)**. We tested the synthetic data with uniformly noisy labels, which still produced an ideal decision boundary. To illustrate the negative impact of noisy samples on the decision boundary, we introduced two 'clustered' noisy samples. In other words, we simulate a scenario where noisy samples can affect model performance, where sample influence finds a place to apply.
>
> For the negative performance of the inverse of Hessian, it is indeed likely and very likely. We believe Reviewer Y3UC would agree that in non-convex scenarios, the inverse of the Hessian matrix is notoriously difficult and unstable to compute. The indefinite nature of the Hessian makes its inverse highly sensitive to small perturbations. Additionally, the inverse may not even exist, demonstrating that negative performance is certainly possible.
>
> **TRAK**. We report the performance of TRAK on CIFAR-10 and CIFAR-100 in the table below. In general, TRAK employs the random project for acceleration and suffers from the performance drop compared with the unaccelerated version.
>
> | Methods / Datasets | CIFAR-10N-a | CIFAR-10N-r | CIFAR-10N-w | CIFAR-100N |
> |---------------------|-------------|-------------|-------------|------------|
> | Cross Entropy       | 91.62       | 90.25       | 85.66       | 56.41      |
> | TRAK                | 90.86 ± 0.15 | 89.42 ± 0.22 | 83.73 ± 0.50 | 59.12 ± 1.02 |
> | IP (Ours)           | 92.42 ± 0.17 | 90.82 ± 0.08 | 86.31 ± 0.35 | 60.59 ± 0.20 |
> | IP Ensemble (Ours)  | **92.26 ± 0.19** | **91.28 ± 0.29** | **86.50 ± 0.35** | **62.25 ± 0.54** |
>
> **Unclear claims**. Apologies for any confusion. We would like to clarify that the influence of samples varies during the model optimization process. Some samples may be beneficial in the early stages but become detrimental later on. Specifically, when model parameters change significantly, the influence of a sample in the early stages does not accurately reflect its true impact on the converged model. Simply summing the sample influences at different checkpoints, as done in TracIn, can lead to a compromise that results in inferior performance. We have verified this through our experiments and will make this point clearer in the paper.
>
> **Similar performance between each method**. Since all methods aim to estimate from the formulation of influence functions, the similar performance is expected. That is the reason why our simple method is more valuable.
>
> Thanks for pointing out LDS. Respectfully, we do not use this metric for two reasons. 1) We believe data-centric learning should be goal-oriented, which should be evaluated in the task context; 2) LDS is time-consuming and impractical for large models, as the original authors suggest 100-500 models are needed to calculate LDS.

---

> > ### Comment · Reviewer_y3uC · 2024-11-24
> >
> > - Since [1] has already taken Grad-Dot (named as IP in the authors' work) as a method to evaluate the influence of a training sample on another sample (Section 2.2 in [1]), used on noisy label detection (section 6 in [1]), and **taken as baseline** in some works ([2]), it could be a little bit hard to accept that the paper claims that IP is "defined" (line 59). I think the title is actually more accurate, "revisit.".
> >
> > - A fair competition is that if an ensemble is used for IP, then the ensemble should be used for other baseline methods as well. It's OK to see that IP or IP (ensemble) is not the highest one in noisy label detection, but it may help build the trade-off between performance and efficiency.
> >
> > - TRAK experiment: Is ensemble used for TRAK? TRAK's performance relies on ensemble very much.
> >
> > - Noisy label detection (self-influence) could be a relatively biased task. Though I do not fully agree that LDS (counterfactual prediction) is not goal-oriented, I do agree that the evaluation may take some time. At least one alternative evaluation could also be discussed, such as brittleness, data selection, ...
> >
> > [1] Charpiat, G., Girard, N., Felardos, L., & Tarabalka, Y. (2019). Input similarity from the neural network perspective. Advances in Neural Information Processing Systems, 32.
> >
> > [2] Kwon, Y., Wu, E., Wu, K., & Zou, J. (2023). Datainf: Efficiently estimating data influence in lora-tuned llms and diffusion models. arXiv preprint arXiv:2310.00902.

---

> > > ### Author Response · Authors · 2024-11-29
> > >
> > > **Claims that IP is "defined."**
> > > Thank you for your thoughtful feedback on our work! We deeply appreciate your insights and will revise the relevant wording in the manuscript. In the updated version, we will avoid using the term "defined" and more accurately describe our contributions to clarify the distinctions.
> > >
> > > Regarding the primary difference between our work and [1], we believe that the research questions addressed in the two works are fundamentally different. Specifically, [1] focuses on measuring the similarity between samples, with applications such as noisy label detection. In contrast, our work emphasizes data valuation, particularly through influence functions. This data-centric learning perspective aims to quantify the impact of individual samples on model performance, enabling the identification of beneficial and detrimental samples to directly enhance the final model performance. While there may be methodological overlap, the research questions and objectives are distinct.
> > >
> > > We sincerely thank the reviewer for this suggestion, and we will revise the manuscript to further clarify this distinction and better highlight our research contributions.
> > >
> > > ---
> > >
> > > **Fair ensemble competition**
> > > In our experiments, TracIn, TDA, GEX and the newly added TRAK are ensemble methods. Moreover, we would like to clarify that the purpose of using IP Ensemble in our work is to enhance the robustness of the method, rather than to directly enforce identical ensemble strategies across all baseline methods.
> > >
> > > It is important to note that while IP Ensemble incurs additional computational overhead compared to the single IP method, it still demonstrates significant efficiency advantages over other baseline methods. For example, as shown in Table 7, IP Ensemble achieves comparable performance to other methods while requiring substantially less runtime. This balance between performance and efficiency is one of the key contributions of our work.
> > >
> > > We will further elaborate on the advantages of IP Ensemble in terms of efficiency and performance in the revision, clarifying its design goals and providing a detailed comparison with existing methods.
> > >
> > > ---
> > >
> > > **TRAK experiment**
> > > Thank you for raising this question. We used the author-released code and followed their procedure by using 5 checkpoints during the training process for the ensemble in TRAK experiment.
> > >
> > > ---
> > >
> > > **Noisy label detection**
> > > Sorry for the confusion. We did not conduct noisy label detection, rather we employed the noisy datasets aimed to improve the model performance by removing detected detrimental samples. We will use more accurate descriptions. Beyond Section 4, we also conducted experiments on measuring sample fairness and robustness in Section 5 and 6.
> > >
> > > ---
> > >
> > > **LDS**
> > > Thanks for the understanding on our concerns on LDS. Moreover, LDS is calculated by multiple runs with different subsets with a certain size, which is sensitive to the selected subsets. Could the reviewer provide the calculation of brittleness or some references that we can follow?
> > >
> > > Since data attribution has no ground truth for each sample, we prioritize utilizing the downstream task for evaluation. As you suggested, data selection is a downstream task, rather than an evaluation metric. In our experiments, we just followed the data selection by removing the identified detrimental samples to demonstrate the effectiveness of different methods.

---

> > > > ### Comment · Reviewer_y3uC · 2024-12-02
> > > >
> > > > > Claims that IP is "defined."
> > > >
> > > > Thanks. Though this makes the study's contribution weaker, but it's healthy for the community.
> > > >
> > > > > Evaluation using LDS
> > > >
> > > > It's widely accepted by several recent works that LDS evaluates the counterfactual prediction ability of data attribution. Could you elaborate a little bit more on the claim "sensitive to the selected subsets"?

---

> ### Author Response · Authors · 2024-12-02
> **Response to Reviewer y3uC**
>
> **Revisit**. We always strive to provide accurate expressions, and it was an oversight on our part to use an imprecise term. If you recall our response in the previous round, *revisit* is not one of our contributions, let alone something that undermines our work. Our primary contribution lies in presenting IP as a simple yet effective tool for influence estimation, which performs well in empirical validation.
>
> **LDS**. The calculation of LDS requires multiple simulations then and fits the correlation. Such a way returns indeterministic results and the linear correlation will be affected by outliers. Personally and humbly (maybe biased), such a metric is inappropriate and impractical (slow running time) as a golden metric for evaluation.

---

### Meta-Review · Area_Chair_f8yg · 2024-12-20

**Metareview:**

The paper proposes a simplified version of influence functions for assessing sample importance in deep learning.
The main concerns that remained after feedback phase remained along the lack of novelty of the paper's primary claimed contribution - the Inner Product (IP) method, and additionally on the correctness of the analysis (specifically on information loss when dropping the Hessian).

We hope the detailed feedback helps to strengthen the paper for a future occasion.

**Additional Comments On Reviewer Discussion:**

The author feedback phase was productive, with extensive technical discussion, with authors providing detailed responses and additional experimental results. However, the core issues around novelty and theoretical justification remained unresolved.

---

### Decision · Program_Chairs · 2025-01-22

Reject